# Fading regulation of diurnal temperature ranges on drought-induced growth loss for drought-tolerant tree species

Xianliang Zhang[1,2], Tim Rademacher [3,4,5], Hongyan Liu [2] ✉, Lu Wang[2] & Rubén D. Manzanedo[6]

Warming-induced droughts caused tree growth loss across the globe, leading to substantial carbon loss to the atmosphere. Drought-induced growth loss, however, can be regulated by changes in diurnal temperature ranges. Here, we investigated long term radial growth responses of 23 widespread distributed tree species from 2327 sites over the world and found that species' drought tolerances were significantly and positively correlated with diurnal temperature range-growth loss relationships for the period 1901-1940. Since 1940, this relationship has continued to fade, likely due to asymmetric day and night warming trends and the species' ability to deal with them. The alleviation of reduced diurnal temperature ranges on drought-induced growth loss was mainly found for drought resistant tree species. Overall, our results highlight the need to carefully consider diurnal temperature ranges and species-specific responses to daytime and nighttime warming to explore tree growth responses to current and future warmer and drier climates.

Global warming is accentuating drought stress worldwide[1–3], leading to tree growth reduction or mortality across forest biomes[4–7], and resulting in a substantial loss of carbon from forests globally[8–10]. Global warming is, however, not necessarily symmetric, and nighttime temperatures seem to be rising quicker than daytime ones, i.e., changing diurnal temperature range (DTR). Nighttime and daytime temperature can also have asymmetric effects on tree growth, especially during droughts[11–13], affecting vegetation growth and ultimately net primary production[14–16]. Reduced DTR has been found to alleviate drought-induced growth loss for larch forests[13]. However, whether this relationship may hold true across tree species is still unknown.

Tree growth mainly occurs at night, when stem turgor is greatest[17,18] while the carbohydrates that fuel photosynthesis are produced at daytime[19]. Consequently, we can expect that changing day and nighttime temperatures independently should have distinct effects on plant carbon dynamics[19–21]. Daytime maximum temperature (Tmax) would likely influence primarily photosynthetic assimilation, while nighttime minimum temperature (Tmin) would mainly affect respiration[19]. Faster day than night time warming (increasing DTR) has been reported in multiple ecosystems at latitudes over 60° N[22]. In lower latitude regions, nighttime temperature increasing faster than daytime temperature (reduced DTR) has been linked to forest ecosystem carbon loss[22,23]. Higher daytime temperature has been suggested to increase vegetation productivity in wet and cool boreal regions, but decrease it in dry temperate regions[22]. However, nighttime warming effects are more ambiguous, as it has been linked to decreasing growth in boreal regions but inconsistent effects in dry temperate zones[24]. Soil moisture may help explain this ambiguity. For example, both daytime and nighttime warming have been shown to reduce tree growth in extremely dry soils[25]. Overall, how changes in diurnal temperature ranges and drought stress interact to affect tree growth remains a critical area of active research.

Asymmetric changes in daytime and nighttime temperatures can have divergent effects on drought-induced growth loss. Daytime

[1]College of Forestry, Hebei Agricultural University, Baoding 071001, China. [2]College of Urban and Environmental Sciences, Peking University, 100871 Beijing, China. [3]Institut des Sciences de la Forêt Tempérée, Université du Québec en Outaouais, Ripon, QC J0V 1V0, Canada. [4]Centre ACER, Saint-Hyacinthe, QC J2S 0B8, Canada. [5]Harvard Forest, Harvard University, Petersham, MA 01366, USA. [6]Plant Ecology, Institute of Integrative Biology, D-USYS, ETH-Zürich, 8006 Zürich, Switzerland. ✉e-mail: lhy@urban.pku.edu.cn

warming—generally leading to higher DTR—can accentuate drought stress on tree growth as excessive daytime temperatures reduce stomatal conductance and photosynthesis[26]. However, nighttime warming—generally leading to lower DTR—seem to have a more complex link with tree growth and drought. Growth chamber experiments have reported enhanced drought stress during nighttime warming[27], while field studies seem to suggest that nocturnal warming alleviates drought stress on tree growth[13,28]. The physiological mechanisms remain unclear: nighttime warming could stimulate respiration, leading to a net carbon loss on the diurnal scale[22], but it may also trigger compensatory photosynthesis during the following day and therefore increase carbon gain[29,30]. Consensus on how and when changing daytime and/or nighttime temperatures alter drought-induced growth loss has remained challenging to achieve. Furthermore, responses to drought stress is likely to differ between species. For example, when percentage loss of hydraulic conductivity, which measures xylem embolism vulnerability, is low, tree species more resistant to drought stress tend to grow less during drought[6]. This discrepancy may be related to, *inter alia*, different stomatal conductivities influencing $CO_2$ assimilation[31], and presumably affecting tree growth, as stomatal response to temperature and drought has been related to species' hydraulic traits[31,32]. While the influence of changing DTR on tree growth during drought is fairly well understood, the linkage between hydraulic traits and the regulation of DTR on drought-induced growth loss is still poorly known.

Here, we used a global tree-ring network from 2327 forest sites and 23 tree species, available via the International Tree-Ring Data Bank (ITRDB) that encompass most temperate and boreal forests in the northern hemisphere. After detrending temporal growth trends to remove non-climate effects (i.e., age, size, etc), site standardized chronologies were used to compare tree growth between regions with high and low DTR under drought and non-drought conditions for the same species. We investigated the relationship between DTR and drought-induced growth loss for each species and obtained species-specific hydraulic traits to test their relevance to explain the interaction between DTR, drought, and growth loss. Temporal changes of DTR effects on drought-induced growth loss from 1901–1980 was quantified using linear mixed models. We hypothesized that species' stem hydraulic vulnerability would positively influence the regulation of DTR on drought-induced tree growth loss (Fig. 1).

## Results

### Influence of DTR on growth during non-drought and drought years

Site's summer DTR had a positive influence on tree growth during non-drought years for 14 species (Fig. 2). However, influences of summer DTR on tree growth was species-specific during dry years (Fig. 2). Tree growth was positively correlated with summer DTR for coniferous tree species such as *Abies lasiocarpa* (ABLA), *Picea engelmannii* (PCEN) and *Tsuga mertensiana (*TSME), while it had a negative influence on tree growth for broadleaf species like *Fagus sylvatica* (FASY), *Quercus alba* (QUAL), *Quercus robur* (QURQ), *Quercus* spp. (QUSP), and *Quercus stellata* (QUST) during dry years (Fig. 2). Positive correlations between DTR and tree growth were mostly concentrated in higher latitudes (Fig. S2), where tree growth in most sites is sensitive to summer temperature (Fig. S3). Negative effects of summer DTR on tree growth were more prevalent in lower latitudes (Fig. S2), where tree growth seemed most limited by summer precipitation (Fig. S3). DTR-growth correlations were positive when growth was primarily related to temperature, but negative when precipitation had a stronger correlation with growth during drought years (Fig. S4).

During non-drought conditions, tree growth indices in high DTR regions were higher than those in low DTR regions for species FASY, *Larix decidua* (LADE), *Larix gmelinii* (LAGM), and PCEN, while they were lower in high DTR regions than low DTR regions for species LASI (Fig. S5). Site mean DTR had a strong negative influence on tree growth indices during drought for FASY, LADE, LAGM, PCEN, *Picea glauca* (PCGL), *Picea mariana* (PCMA), *Pseudotsuga menziesii* (PSME). However, we found that this negative relationship during dry years turned positive during normal and wet years. Tree growth indices in low DTR regions were generally higher than those in high DTR regions during drought conditions for tree species FASY, LADE, LAGM, and PCEN (Fig. S5). By contrast, tree growth indices of LASI and QUAL were higher in high DTR regions than in low DTR regions during droughts. In general, site mean DTR had opposite effects on tree growth indices during dry and non-drought years. However, DTR showed non-

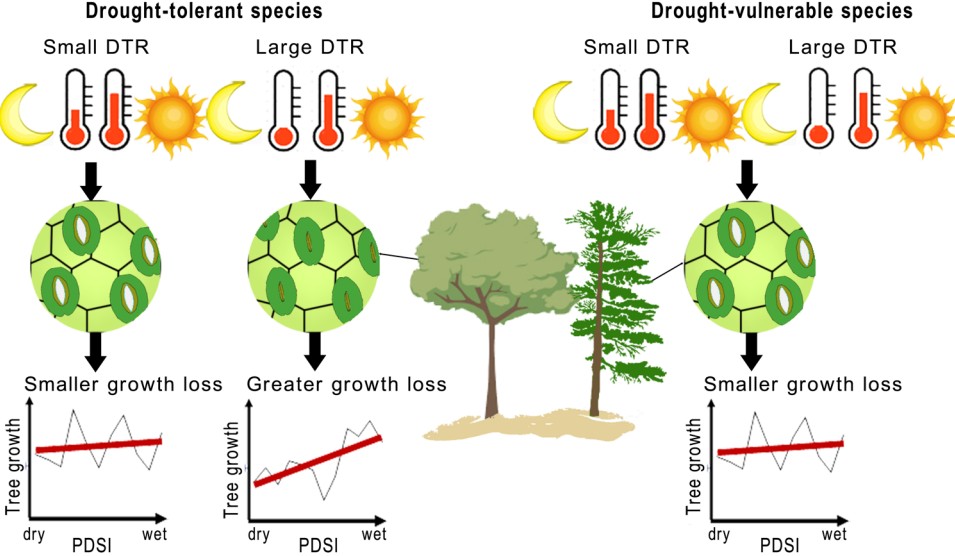

**Fig. 1 | A schematic diagram of how DTR regulates drought-induced growth loss for drought-vulnerable and drought-tolerant species.** Growth loss was defined as the percentage of growth reduction from non-drought growth to drought growth. The stomates of drought-vulnerable species are more sensitive to droughts, trees have greater growth losses in regions with higher DTR while smaller growth loss were observed in region with lower DTR due to the presumed regulation of DTR on drought-induced growth loss. In contrast, the stomates of drought-tolerant species are not sensitive to droughts, with trees only experiencing small growth loss during droughts.

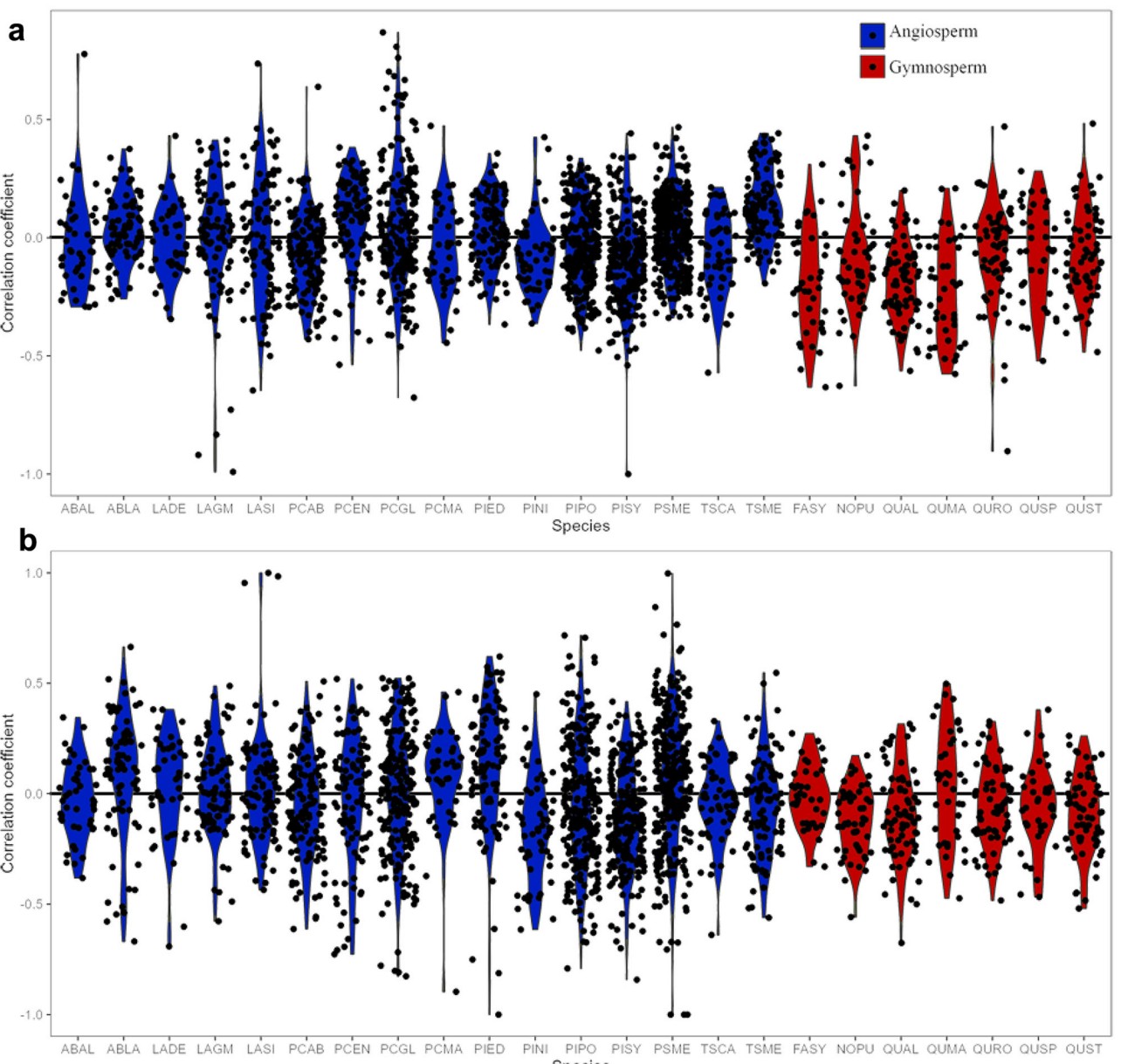

**Fig. 2 | Species-specific relationships between DTR and tree growth.** Partial correlation between DTR and growth considering influences of temperature during dry (**a**) and wet (**b**) years for each species. Species IDs can be found in Table S1. Black dots represent site correlation coefficients with colored violin plots showing the distribution of these for each species.

significant influences on tree growth for more than 10 species during both dry and non-drought years.

**Regulation of hydraulic traits on DTR-growth loss relationship**

Sites with higher mean Tmax showed higher growth losses, whereas high Tmin alleviated drought-induced growth loss, indicating that site mean DTR contributed to drought-induced growth loss via either Tmax or Tmin (Fig. 3). Parallel Random Forest models similarly indicated that site mean DTR was an important determinant of drought-induced growth loss (Fig. S6). Correlation analysis showed that site mean DTR had a strong positive influence on drought-induced growth loss for species LADE, LAGM, PCEN, PCGL, PCMA, PSME and QUST (Fig. S7), where higher growth loss was recorded in regions with higher DTR during dry years. Drought-induced growth loss and site mean DTR were negatively correlated for the species

QUAL, LASI, and QUMA during dry years and the remaining species showed no clear relationship between drought-induced growth loss and site mean DTR.

The influence of DTR on drought-induced growth loss was strongly positively correlated ($R^2 = 61\%$, $p < 0.05$) with lethal water potential (P50 for gymnosperms and P88 for angiosperms, Fig. 4), which decreased with drought tolerance. Site mean DTR had a significant and positive impacts on drought-induced growth loss for species with lethal water potential lower than −3.5 MPa. However, drought-induced growth loss was higher in low DTR regions for species with lethal water potential higher than −3.0 MPa. Site mean DTR had limited effects on drought-induced growth loss for species with intermediate lethal water potential −4 to −3 MPa. Whether drought-induced growth loss was influenced by site mean DTR was dependent on the species' drought tolerance.

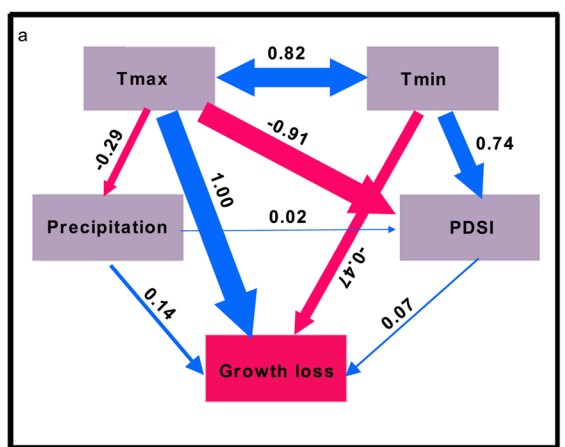
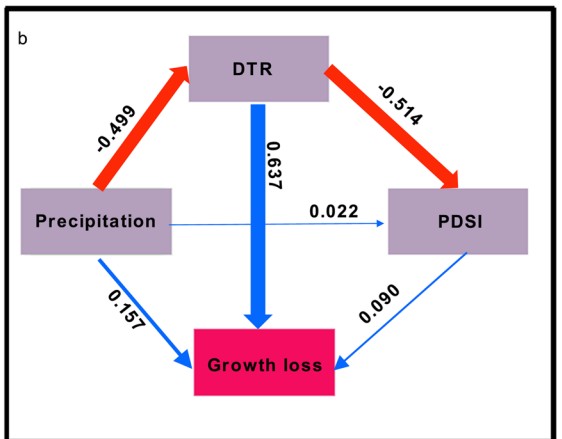

**Fig. 3 | Relationship between climate variables and growth loss identified from structural equation models.** Two models that included either Tmax and Tmin (**a**) or DTR (**b**) were developed to detect the influences of Tmax and Tmin or DTR on growth loss. Red lines indicate negative influences while blue lines represent positive influences.

**Fig. 4 | Relationship between drought tolerance measured by lethal water potential (i.e., $P_{50}$ for gymnosperms and $P_{88}$ for angiosperms) and the correlation coefficient between drought-induced growth loss and DTR.** Species ID can be found in Table S1. Low lethal water potential indicates high drought tolerance. Dots mark mean correlation and error bars represent the standard error of correlation. Different colors represent different species, the colors are same as in Fig. 6. The line was fitted using two-sided Deming regression.

## Weakening effect of hydraulic traits on DTR-growth loss relationship

The correlation between DTR and drought-induced growth loss increased over time for most species (Fig. S8). Linear mixed models supported this finding by showing that the influence of DTR on drought-induced growth loss increased over time (Table S2). The influence of monthly Tmax and Tmin on drought-induced growth loss seem to have become increasingly positive from 1901 to 1980. Monthly

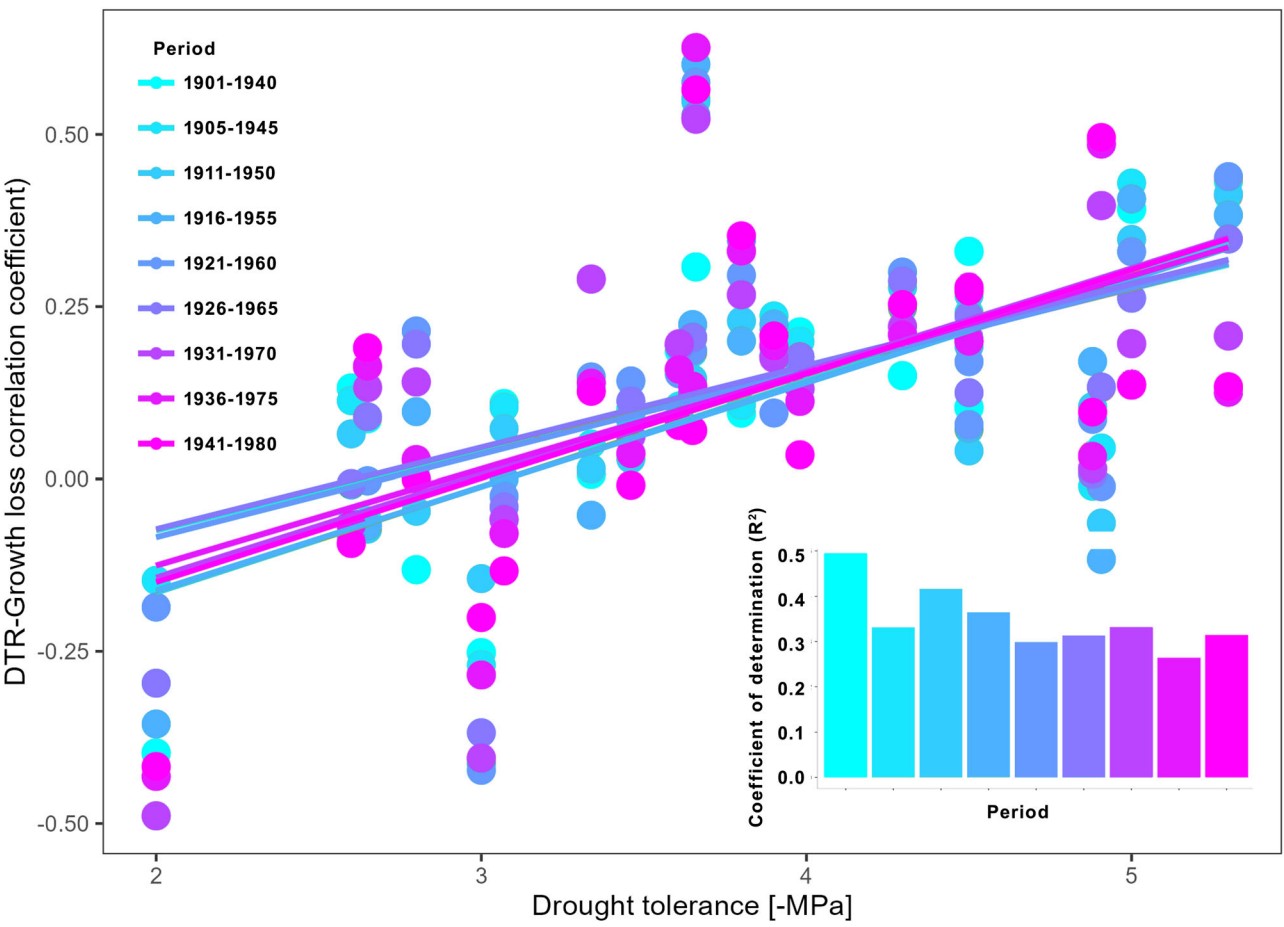

**Fig. 5 | Temporal changes in the relationship between drought tolerance and DTR-growth loss correlation.** Coefficient of determination ($R^2$) is for the linear regression line.

Tmax also had increasingly positive correlations with drought-induced growth loss, while originally weaker correlations between Tmin and growth loss turned negative over time. Temporal changes in DTR-growth loss correlations were negatively influenced by the warming rate of Tmin and Tmax (Fig. S9). We would thus expect that accelerated warming of Tmin and Tmax would lead to a decrease in DTR-growth loss correlations.

The regulation of hydraulic traits on how DTR influences drought-induced growth loss differed over time (Fig. 5). Drought tolerance had a stronger influence on the correlation of DTR and drought-induced growth loss during the period 1901–1940. However, correlation between DTR and drought-induced growth loss started to decouple from species' drought tolerance from the period 1906–1945 to the period 1941–1980. The correlation between drought tolerance and DTR-growth loss got weaker over time. Alleviation of reduced DTR on drought-induced growth was mainly shown for drought-tolerant species (Fig. 4). Such a weakening relationship over time would lead to weaken regulation of DTR on drought-induced growth loss for drought-tolerant species.

## Discussion
### Drought-induced growth loss related to DTR
Our results show that DTR exerts opposite effects on tree growth between regions for many widely distributed tree species, including dominant species of some biomes (e.g., PISY, PSME, and LASI). While previous studies revealed that day and night-time warming had asymmetric influences on tree growth[12,24,25,33], our finding indicates that the influences of shifting DTR on tree' drought-growth responses are diverse across tree species. Larger DTR positively influence tree growth during droughts for coniferous tree species, ABAL, PCEN, and TSME at the site-level, implying that high DTR benefits tree growth during droughts across sites. High DTR could be related to available moisture/humidity as DTR was negatively correlated with precipitation. However, the influence of droughts on tree growth was enhanced by increasing site mean DTR for these species, with droughts more severely affecting tree growth in regions with high mean DTR than in regions with low mean DTR. This indicates that for these coniferous species, high DTR may be beneficial to maintaining tree growth during droughts in regions with low mean DTR, but not in regions with high mean DTR. By contrast, DTR have consistent negative influence on tree growth during droughts for broadleaf tree species (FASY, QUAL, QURQ, QUSP, and QUST). Previous studies reported higher drought resistance in gymnosperm than angiosperm[34], and our results further reveal that this high drought resistance of gymnosperm is particularly pronounced in regions with low DTR.

High daytime temperature, led to higher growth loss during drought[12,33]. In fact, warm daytime temperatures had a stronger influence on tree growth during droughts than warm nighttime temperatures. High daytime temperatures may lead to drought-induced tree growth loss due to reduced stomatal conductance and photosynthesis at particularly high temperatures[35,36]. However, there is still little consensus on how nighttime warming influences tree growth during droughts, with the effects of drought stress on tree growth during nighttime warming likely to be highly species-specific[29]. Our findings show that the drought-induced growth loss was lower in regions with low mean DTR, which is consistent with nighttime warming generally

alleviating drought stress in these regions[13]. Since tree growth mainly occurs at night[17], warm nights increase nighttime respiration, leading to photosynthesis compensation in the next daytime[19], and stimulating carbon sequestration to enhance community resistance to drought[37]. Our results suggested that warmer nighttime temperatures led to an increase in photosynthesis above and beyond what was spent in respiration. The link between wider tree rings and warmer nights reveals a potentially strong compensation between rates and durations of cell differentiation processes mitigated drought stresses on tree-ring structure[29]. It is also possible that wider rings stem from warmer nights and increased snowmelt[38] in areas where growth is limited by snowpack and/or growth limited by cooler night time temperature, such as in the northern boreal forests. Furthermore, the leaf-on period and potentially the growing season length may also be affected by asymmetric warming. Summer vegetation greenness was enhanced by nighttime warming, but decreased by daytime warming in other studies[39]. Nighttime warming led to earlier spring phenology than daytime warming, which prolonged the length of growing season[40]. A combination of compensations in photosynthesis and cell differentiation processes, direct temperature limitation during cold nights, and a prolonged growing season may explain why warmer nights alleviate drought stress in regions with low DTR. However, photosynthesis and growth are often decoupled, particularly during drought[41–44]. These may also be the reasons why drought-induced growth loss was higher in regions with high DTR.

It should be noted that the modulation of DTR on drought-induced growth loss was not evident for all widely distributed tree species. The alleviating effect of DTR on drought-induced growth loss was stronger in drought-tolerant species. Tree species with low xylem hydraulic conductance (xylem pressure at which 50% of conductivity is lost ($P_{50}$) for gymnosperm and 88% ($P_{88}$) for angiosperm) exhibited higher resistance to droughts[6]. High hydraulic safety margins generally correspond to lower stomatal conductance following a well-established tradeoff between hydraulic safety and efficiency[45–47]. High DTR can exacerbate drought conditions and has been shown to lead to low stomatal conductance during droughts[48] and reduced growth eventually[11,33]. However, stomatal conductance was not sensitive to drought conditions for drought-vulnerable species compared to drought-resistant species[45]. Therefore, the drought-induced growth loss was not sensitive to DTR for drought-vulnerable species.

### Decoupling of relationship between hydraulic traits and DTR-growth loss relationship

Our results show that the warming rates of minimum and maximum temperatures influence DTR-growth loss relationships. Asymmetric diurnal warming, with more pronounced nighttime warming has been observed already in large parts of the Northern Hemisphere, leading to a large scale decrease in DTR[11]. The weaker correlation of drought-induced growth loss and DTR indicates that drought-induced growth loss reduced with increasing DTR. This confirmed that warmer nighttime temperatures reduced the growth loss caused by drought not only for larch, as it has been previously established[13], but across many other widely distributed species worldwide. Increased nighttime temperatures enhanced nighttime respiration, and consume stored non-structural carbon (NSC) while enhancing tree growth, which mainly occurred at night[17]. Although greater growth loss had been reported due to carbon starvation during severe droughts, increasing nighttime temperatures would be beneficial to tree growth during mild to severe droughts when neither temperature nor NSCs are a major constraint for tree growth. However, daytime warming (e.g., by increasing maximum temperature) has been reported to accelerate drought stress, leading to greater growth loss during droughts[12]. While maximum temperature and minimum temperature had opposite effects on drought-induced growth loss, increased DTR weakened the

influence of DTR on drought-induced growth loss. It is important not only to consider the direct and indirect effect of maximum and minimum temperatures (drought and frost tolerances), but also to consider how changes in these may influence tree growth via their indirect effect on experienced temperature ranges.

Our results show that the relationship between DTR and growth loss has started to be increasingly decoupled from hydraulic traits since 1940. In the earliest study period 1901–1940, low DTR seemed to be beneficial to tree growth during droughts for drought-tolerant species, but did not influence the growth of drought-vulnerable species as much. However, drought-induced growth loss was reduced in recent decades for these drought-resistance species, which was strongly related to reduced DTR. Therefore, although trees have high hydraulic conductivity and reduced stomatal conductance during droughts, which eventually reduces photosynthetic $CO_2$ assimilation, reduced DTR had been reported to alleviate growth loss for drought-resistant species[13]. This likely explained why we observed a weakened relationship between lethal water potential and the correlation between DTR and growth loss during dry years. However, it should be noted that multi-species studies showed that higher DTR can reduce the abundance of dominant, stable species, and lower community temporal stability[37]. Therefore, increasing asymmetric diurnal warming may affect species distributions and community dynamics, especially under changing drought regimes.

In summary, DTR had contrasting influence on tree growth during dry and normal/wet years across sites. Drought-induced growth losses were strongly influenced by DTR for drought-tolerant species. However, this relationship has faded after 1940, likely due to a weakening relationship between DTR and growth loss caused by beneficial nighttime warming. Our results indicated that the regulation of DTR on drought-induced growth loss has changed with asymmetric day and night warming.

## Methods
### Data synthesis
**Tree-ring data.** Tree-ring width data were retrieved from the International Tree-Ring Data Bank (https://www1.ncdc.noaa.gov/pub/data/paleo/treering/, accessed July 1 2022). We selected widely distributed, canopy dominant tree species with enough sample depth in the ITRDB according to the following criteria: (1) raw ring width data for individual trees was available for the study period 1901–1980 and (2) there was data for at least 40 sites for the species, once we removed all sites with no tree above 140 years of age, as young trees are known to be more sensitive to droughts[49]. In total, 23 species from 2327 sites were used in the following analyses. It should be noted that some oak species were lumped as *Quercus spp.* (QUSP) in the ITRDB. We ended up using only raw tree-ring data of the period 1901–1980 to ensure sufficient temporal replication, because after 1980 sample depth rapidly dropped (Fig. S1). Standardization was conducted using the dplR package (v.1.6.8)[50] within the R programming environment (R Core Team 2021), to remove age-related biological trends for every site. Site chronologies were developed using the Friedman detrending method with alpha = 5[51], a widely used detrending approach in dendroecological studies. Of the 23 target species, 16 were gymnosperms and 8 angiosperms (Fig. 6). This, though unbalanced, improves upon the high imbalance toward gymnosperms reported in the ITRDB[52]. Our data was also clearly shifted toward northern European and north American forests, which again is a known problem of the database[52] that may influence our results (but see ref. 53). Most ITRDB samples are selected for particular climate sensitivity, suggesting that they are overly sensitive compared to an ecological sampling network[54], thus any inferred drought sensitivity/drought loss might also be overestimated. The most abundant species in our data were *Pseudotsuga menziesii* (PSME, 317 sites), *Pinus sylvestris* (PISY, 245 sites), and *Pinus ponderosa* (PIPO, 217 sites) (Fig. 6 and Table S1).

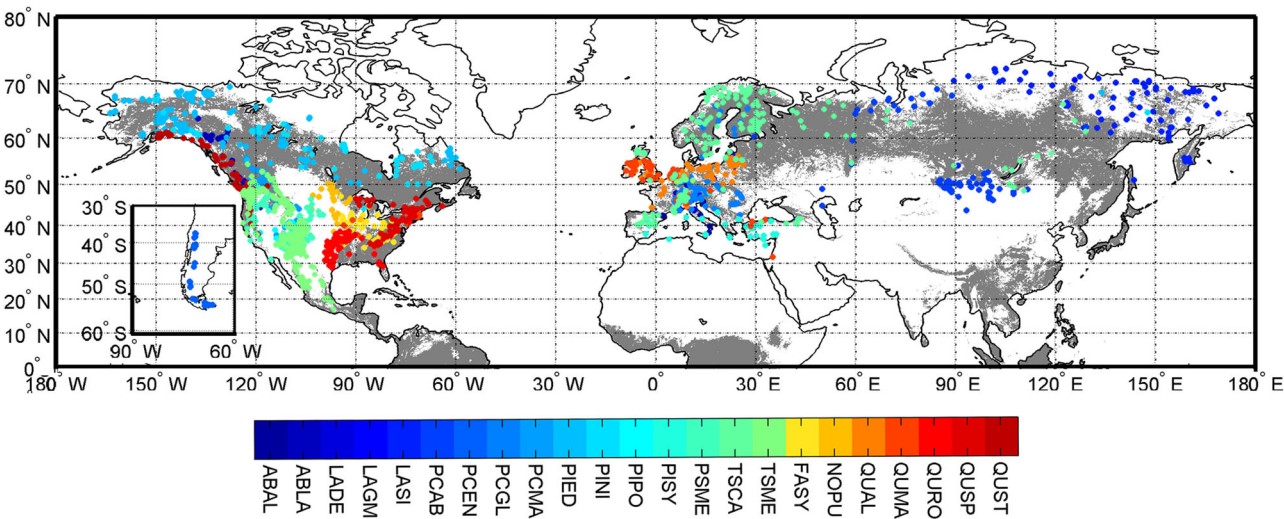

**Fig. 6 | Distribution of 23 tree species retrieved from ITRDB.** The gray regions represent forest regions, and the white regions represent non-forest regions. For the full name of each species refer to Table S1. The blue colors are gymnosperms and red colors are angiosperms.

**Climate metrics.** Drought intensity was evaluated using the Self-calibration Palmer Drought Severity Index (scPDSI)[55]. Monthly gridded global scPDSI, maximum, mean, and minimum temperature, as well as monthly precipitation with a spatial resolution of 0.5° from 1901 to 2015 were obtained from the CRU v4.05 dataset (https://catalogue. ceda.ac.uk/uuid/3f8944800cc48e1cbc29a5ee12d8542d) for the grid cells containing every site's location. Monthly DTR was calculated by subtracting the monthly maximum temperature from the monthly minimum temperature.

**Species' drought tolerance.** Lethal water potential represents a species' drought tolerance. Lethal water potential is correlated with xylem pressure at which 50% of conductivity is lost ($P_{50}$) for gymnosperm and xylem pressure at which 88% of conductivity is lost ($P_{88}$) for angiosperm[6,56]. Low lethal water potential indicates high drought resistance for a tree species. P50 for gymnosperm, P88 for angiosperm and hydraulic safety margin (HSM, defined as differences between naturally occurring xylem pressures and pressures that would cause hydraulic dysfunction) were retrieved from Choat et al.[34] for most species. For those species not listed in Choat et al.[34], $P_{50}$ and $P_{88}$ values were obtained from the literature (Table S1) except for *Tsuga mertensiana* (TSME) and *Nothofagus pumilio* (NOPU) for which we found no P50 value. There is variation around $P_{50}$ and $P_{88}$ values for different populations of species, however, we assume hydraulic traits for species are relatively constant within the same trees/population over time.

**Statistical analysis**
**Factors influencing tree growth.** Correlations between site tree-ring chronologies and site climate (i.e., temperature and precipitation) were calculated to detect the dominant climate factor for each site. We also calculated correlations between site summer (June–August) DTR and tree-ring index for each site to evaluate the influence of summer DTR on tree growth during relatively dry (summer mean PDSI < −0.5) and wet (summer mean PDSI > 0.5) conditions.

Site-level mean summer DTR was calculated as the mean DTR value over the period 1901–1980 for each site. Mean tree growth indices during droughts were calculated by averaging ring width indices of dry years (PDSI < −0.5), while growth indices under non-drought conditions were calculated by averaging ring width indices over wet years (PDSI > 0.5). To investigate the variation of mean tree growth indices with site mean DTR during dry and wet years, across

sites and species, we calculated correlations between site mean DTR and mean tree ring indices during dry years and wet years.

**Relationship between DTR and drought-related growth loss.** Drought-related growth loss per site was then calculated according to Au et al.[49] as follows:

$$\text{Growth}_{\text{loss}} = \frac{\overline{\text{Growth}_{\text{non−drought}}} - \overline{\text{Growth}_{\text{drought}}}}{\overline{\text{Growth}_{\text{non−drought}}}} \quad (1)$$

where $\text{Growth}_{\text{loss}}$ is the drought-induced growth loss, $\overline{\text{Growth}_{\text{non−drought}}}$ the mean tree growth indices in non-drought years (4 > PDSI > 0.5), and $\overline{\text{Growth}_{\text{drought}}}$ is the mean tree growth indices in years with mild to severe drought (−4 < PDSI < −0.5). We excluded severe drought years (PDSI < −4), as high DTR has been found to not alleviate tree growth under severe drought[13].

The interactions of climate variables and drought-induced growth loss were identified with structural equation models. DTR is highly correlated with Tmax and Tmin (Fig. S10), therefore, two models that included either DTR or Tmax and Tmin were developed (Fig. 3). Variables used in one model were drought-induced growth loss for the 23 species, site mean summer DTR, precipitation and PDSI. Whereas drought-induced growth loss for the 23 species, site mean summer Tmax, Tmin, precipitation and PDSI was used as variables in another model. Models were tested and the final model with the best fitness index was used ($\chi^2$, $p$ value, normed fit index (NFI), comparative fit index (CFI), and root mean square error of approximation (RMSEA), with NFI > 0.9, CFI > 0.9, $p$ > 0.05 and, lower $\chi^2$ and RMSEA indicate satisfactory fit).

**Importance of drivers of drought-induced growth loss.** The relative importance of each climate variable with regards to growth loss was detected using the Boosting Regression Tree (BRT) method. The linear relationship between site mean DTR and drought-induced growth loss was analyzed using errors-in-variables correlation for each species. Since the number of sampling sites were not the same for different species, we randomly selected 50 sites with no replacement to calculate the correlation between DTR and drought-induced growth loss to ensure the degrees of freedom were consistent across species. The standard error and mean correlation were calculated from 1000 runs.

**Demming regression of DTR regulation and function traits.** Functional trait relationships with the associations between site mean DTR and drought-related growth loss were analyzed using Deming regressions which fit a straight line to two-dimensional data where both variables, $X$ and $Y$, are measured with their respective errors. We used $P_{50}$ as a proxy for drought tolerance for gymnosperm and $P_{88}$ for angiosperm. The Deming regressions were conducted using the "deming" package in R[57].

**Moving interval response analysis of the correlation of DTR and growth loss.** Temporal changes in correlation between DTR and growth loss were investigated using moving interval response analysis in the period 1901–1980. This 80-year period is likely sufficient to detect strong trends potentially induced by climate change. The variation of bootstrapped correlation coefficients between DTR and growth loss was assessed using moving interval response analysis.

**Changing contribution of climate variables on drought-induced growth loss.** We quantified changing contributions of DTR, PDSI, Tmax and Tmin to growth loss using linear mixed-effects models. In these analyses, DTR, PDSI, Tmax and Tmin were included as fixed effects, while age and species were included as random effects. Temporal changes in the contribution of DTR, PDSI, Tmax and Tmin to growth loss were obtained by comparing linear mixed-effects models fitted to different periods.

**Changing relationship between warming rate and DTR-growth loss correlations.** Linear trends of changing DTR-growth loss correlations over time were calculated for each species. Correlation coefficients between the linear trend of DTR-growth loss correlation and warming rate of maximum and minimum temperatures were calculated to detect the influence of changing maximum and minimum temperatures on the changes of DTR-growth loss relationship.

**Changing relationship between hydraulic traits and DTR-growth loss correlation.** Species hydraulic traits were assumed to be constant over time, while the DTR-growth loss correlation changed. Temporal changes in the hydraulic traits and climate-growth loss correlation were then calculated to investigate changes in the regulations of hydraulic traits on the DTR-growth loss correlation for the period 1901–1980.

For all moving window analyses the interval was fixed at 40 years with 5-year steps beginning with 1901–1940 and ending 1951–1980. The sensitivity of changing window size was tested by changing the window size from 30 to 50 years, which did not change the results qualitatively (Fig. S11).

**Reporting summary**
Further information on research design is available in the Nature Portfolio Reporting Summary linked to this article.

## Data availability
Tree-ring width data were retrieved from the International Tree-Ring Data Bank (https://www1.ncdc.noaa.gov/pub/data/paleo/treering/). Monthly gridded global scPDSI, maximum, mean, and minimum temperature, as well as monthly precipitation were obtained from the CRU v4.05 dataset (https://catalogue.ceda.ac.uk/uuid/3f8944800cc48e1cbc29a5ee12d8542d).

## Code availability
The codes used to calculate the results reported in this study have been deposited on GitHub: https://github.com/zhxianliang/Fading-regulation-of-dtr-on-drought-induced-growth-loss.

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

## Acknowledgements

H.L. received support from State Key Research and Development Program (2022YFF0801803) and Natural Science Foundation of China (42161144008). X.Z. received support from S&T Program of Hebei (226Z6801G), and Talent introduction program in Hebei Agricultural University (YJ201918). We thank all contributors to the ITRDB to make this analysis feasible.

## Author contributions

H.L. and X.Z. conceived the study. H.L. supervised and administered the project. X.Z. designed the experiments, analyzed the data, and wrote the original draft. T.R., R.D.M., H.L. and L.W. reviewed and edited the draft. All authors read and reviewed the manuscript.

## Competing interests

The authors declare no competing interests.
