## [Peer Review File · Nature Communications]

Fading regulation of diurnal temperature ranges on drought-induced growth loss for drought-tolerant tree speciesREVIEWER COMMENTS

Reviewer #1 (Remarks to the Author):

Review of Fading regulation of diurnal temperature ranges on drought induced growth loss

Comments for Author

Overall goal: Use ITRDB to evaluate how DTR affects species-level responses to drought (drought-induced growth loss) across space, and time.

Overall summary:

The authors leverage a large, multi-species database of tree ring time series data to assess the effect of Diurnal temperature range on drought induced growth loss.

Because they looked at several different taxa across regions that are generally controlled by different drivers (temperature vs precipitation), they find that the effects of DTR on relative tree growth vary widely across species, under different drought vs non-drought conditions, and across space. The authors then use several analyses to try to distill these differences into some common findings.

Noteworthy results & Significance: Quantifying the relationships between physiological drought tolerance parameters and observed DTR-growth loss correlation coefficient is a unique, possibly new approach to link observed growth to physiological metrics. The findings of this work have relevance to predicting responses of trees to future changes in DTR, and in understanding past climates.

This manuscript has several strengths, and the authors research questions are both relevant and timely. Specifically, the key strengths and noteworthy results of this manuscript are:

1. Leveraging a wide network of tree ring data available should allow the authors to quantify differences in the effects of DTR across species, and say something about how it varies across space.
2. The effects of Diurnal Temperature Ranges on drought response are an important topic needed to address how asymmetric changes in high vs. low temperatures could enhance or alleviate negative effects of drought stress.
3. Quantifying the relationships between physiological drought tolerance parameters and observed DTR-growth loss correlation coefficient is a unique, possibly new approach to link observed growth to physiological metrics.

However, there are some methodology, organization, and clarity issues that I believe need to be addressed with this manuscript in revision:

1. The analyses focus on DTR and its effect on growth loss during drought, but don't directly quantify the influence of high summer temperatures in driving high DTR, nor does it directly address the relationships between DTR, tmin, and tmax in the dataset. Based on the maps in the supplement, it does seem that the role of DTR likely depends on whether the tree is in a moisture limited or temperature limited system, or at least that DTR, tmin, tmax, and precipitation are linked. I am not totally convinced that the relationships between DTR and growth loss are not a product of VPD or humidity/ snowpack interactions. I wonder if an initial exploration into drivers (or correlations) of DTR with other climate variables could help shape the mechanisms discussed in the discussion.

Suggestion: Though authors do discuss the role of min vs max temperatures and potential effects of increasing DTR on growing season length, I think that the authors should show the relationships between DTR, tmin, and tmax themselves somewhere in the supplement, as some of the statistical analyses include all three of these variables, which may not be appropriate if they are strongly correlated.

2. The methods section outlines a lot of different statistical analyses, but it is hard to follow what each test is doing, and why you need to report all of these. It is also a little unclear in the methods which values (means, temporal variations, correlation coefficients, individual growth indices or whole chronologies?) are being used and why. This is an issue because it is difficult to evaluate

whether the figure/statistical test is necessary, but also because it makes the manuscript hard to follow.

Suggestion: I appreciate that trying to distill all the complexity in the relationship between DTR and growth loss is a heavy lift, but I think some clarity and organization in the methods section details would help—see statistical analysis comments below. Also, I believe that adding in some key sub headers that are common throughout the discussion, results, and methods would greatly improve the clarity—You already have the start to this in the Results and Discussion, but I would argue for a few more specific sub headers that describe the single question being addressed with each statistical test/analysis, and to use these same ones to organize the methods section. I also wonder whether all these different analyses are needed in this manuscript.

3. I think there is a fundamental issue with using detrended growth indices in some (not necessarily all) of the analyses in this paper. Detrended growth indices (whether at the individual tree, or site-chronology), are detrended to be relative growth for that site. So, growth losses are also relative to that site. So, after detrending, if two sites both have a growth loss of 0.5, they could have drastically different absolute changes in growth. There is a good deal of discussion of growth loss, and how growth losses vary across high vs low DTR regions, but you can't really compare absolute growth losses from these detrended time series. These comparisons are discussed (briefly) in the context of changes in Carbon uptake, which is not something you can quantify with growth indices. Additionally in most places in the manuscript "tree growth" should be changed to "relative tree growth," to distinguish that this analysis was done on detrended data.

4. Issue: The trees in the ITRDB are biased towards highly sensitive sites or individuals, compared to a more ecologically sampled dataset (Klesse et al 2018). Thus, they are likely biased in their relationship to DTR.

Suggestion: At the very least these biases should be acknowledged in the manuscript and in the methods.

5. Issue: Temporal analyses go up to the year 1990, even though the sample depth decreases rapidly after 1980, making the last 2-3 time periods used in these analyses potentially biased by the species and # of individuals present in the dataset.

Suggestion: This either should be discussed/addressed in the discussion, or the authors should limit the time series analysis to 1900-1980.

Relating to issue #2 above, the following are different statistical analyses that I gathered from the methods section, with a few questions about why these statistical methods are used.

1. Correlation of annual growth & annual climate variables
2. Linear regression of annual (?) summer DTR and site-level chronologies, in wet and dry years
3. Linear regression of average site DTR and average growth in wet and dry years
4. Boosting regression tree to identify importance of drivers of drought induced growth loss.
5. Demming regression of DTR and function traits
6. Moving interval response analysis of correlation of DTR & growth loss
7. Random Forest Model to detect shifts in contribution of climate variables.
8. Linear mixed effects models to quantify changing Contribution of climate variables to growth loss.
9. Linear trends fitted to changes in DTR-growth loss correlations from moving interval response analysis.
10. Correlation between linear trend of DTR growth loss and warming rate
11. Calculated temporal changes in the hydraulic traits and climate growth loss correlation are calculated.

Why are analyses 1-4 not combined into single linear regression that predicts annual growth (or annual growth loss), from annual DTR, temperature, precipitation, site-level DTR, species, and other covariates used in the boosting regression trees with random slopes for wet and dry years or even species?

It's also not clear from the methods and results why both a random forest model and a linear mixed effects models were used to quantify changes in contribution of climate variables to growth loss. The linear trends fitted to the DTR-growth loss correlations are later used to compare to the rate of warming, so these two make sense and are presented in the results, but I wonder why you did also relate them to Tmax?

You also say that you calculated temporal changes in hydraulic traits and climate growth loss correlation, which is maybe referring to the results in figure 7 (?), but it was difficult to tell, since the authors don't provide enough information to reproduce these analyses.

General line by line comments:

Line 26: Use of regulation here does not state the direction...is it consistently negative? Also asymmetric day and night warming does not necessary state the direction either. Are days or nights heating faster consistently?

Line 40: Litter should be little.

Line 39: I think there are a few other studies on changing DTR effects on other metrics of growth—NPP possibly?

Line 53: Is dryness here a soil property/characteristic? Or a state of soil moisture. At the beginning of this sentence, I thought you were going to talk about soil properties (%sand, soil type, etc.)

Line 54: I think this contradicts the statements in previous lines—51-52.

Line 61: stomatal*

Line 63: enhance should be enhanced.

Line 64: This is missing a word somewhere—field studies supported the hypothesis?

Line 67-69: Does compensatory photosynthesis actually increase carbon gain? It would have to compensate for the nighttime respiration, and then be enhanced/primed to go above and beyond what was lost in respiration.

Line 70. You should introduce the % loss of hydraulic conductivity a little more here.. Is this typically a species-wide parameter or measured over time or site conditions?

Line 80-83: Are you using Individual tree responses?

OR site level chronologies? It sounds like individuals from this sentence.

lines 83-85: This sentence is good, but I am wondering if you can change this sentence slightly or add a sentence that briefly overviews how you did this—that is outline the different analyses that you used to investigate the relationship between DTR and drought induced growth loss for each species. This would help with following the methods and results, since there are several analyses that involve correlations with DTR. You also don't mention looking at the weakening DTR temporal effects at all here, but it should be included here as it is in the title of the paper.

Lines 91-95: Most of these correlation distributions are overlapping with zero according to the figure. Could you define the threshold at which you determined these are positive or negative correlations. Also you should say what the significance values are testing in the figure. I think it is differences within species in dry and wet years.

Line 94: Is is site-level DTR, or annually varying? I think if it is correlations in dry years then it is annually varying?

Line 96: spelling error—should be *Quercus* spp.

Also, Perhaps you should explain in the methods why you lump some quercus all together in *Quercus* spp, and separate others. I think it has to do with the structure of the data in the

database, but I am not sure.

Line 105: you spell out all the scientific names except FASY here.

Lines 108-115: You are comparing growth indices, not growth here, so make sure to state that. Because the data is detrended by site, its hard to actually say anything about growth across space.

Lines 118-119: so this is the mean DTR vs mean growth induced growth loss over time for all chronologies of that species? individuals?

Line 122: What about LASI, QUMA? These also look negative in figure S7.

Lines 131-135: How much variance does p50 or p88 explain in the relationship?

Lines 140-142: Why use a random forest model to detect changes, and then a linear effects model? Why not include the temporal component in the regression model?

Line 147: "relative" should be "relatively"

Also the effect of PDSI is the strongest over time, but does also increase according to figure 4. These variables (PDSI and DTR) are likely correlated, along with Tmin and Tmax...how do you parse these effects?

Line 153-154: This is interesting...is it because maximum temperature is also increasing rapidly? Or is there an interplay between VPD/moisture availability and DTR/min temperature that is not looked at?

Line 176: This makes sense...lower temps at night mean some relief from high temps

Line 176-179: There is quite a lot packed into this sentence. I think there should be a comma after tree species on line 177.

Line 183: One question that keeps coming up for me while reading this is: how do you parse the effects of high DTR from the effects of high temperatures during drought? They are linked, right? Okay this gets acknowledged around line 189, but DTR is both high daytime temperature and low night time temperatures.

Line 193..."what refers to..." doesn't really fit what this sentence is trying to say, I don't think.

Line 199-202: Its not clear how increasing respiration would alleviate drought stress. If photosynthesis is compensating for increased respiration, its just a net zero, right? Unless nighttime warmer temperatures lead to an increase in photosynthesis above and beyond what was spent in respiration, or the normal amount of photosynthesis??

Lines 201: Is it also possible that the wider rings stem from warmer nights, and increased snowpack melt in areas where growth is limited by snowpack?

Lines 207-208: High DTR could also be related to available moisture/humidity. For example drier desert areas have larger DTR's, but because of their low humidity.

Is it compensation, or is growth limited by cooler night time temps in some places? Or more likely, since you are looking at yearly variables...its a longer growing season as stated in 207-210.

212-213: this sentence could be stronger if you say the direction of the influence of DTR.

line 216, and this occurs elsewhere. The use of past tense to describe results of previous studies, but without distinguishing it from the results of this study in some way. For this sentence you could say high hydraulic safety generally corresponds to lower stomatal conductance (cite), following a well established tradeoff...

Line 218-219: But is this just because of really high temps?

Line 223: This section title is confusing. Is the hydraulic trait regulation weakened? Or is it the relationship between DTR and growth loss.

Line 269-286: Another issue with the database to highlight is that most ITRDB samples are selected for particular climate sensitivity, suggesting that they are overly sensitive compared to an ecological sampling network (Klesse et al 2018.). This isn't a dealbreaker for this study, but you should point out that any inferred drought sensitivity/drought loss might also be overestimated here.

Line 295: This sentence is a little confusing for me.

Line 297: define psi 88 too. Its implied but good to define it

Line 299: Define hydraulic safety margin too.

Line 306: Why Summer DTR? How is summer defined across all these sites?

Line 309: So there is site-level DTR and a site summer DTR?

Line 310-311: Wondering why the averages are being compared here...
Wondering why the averages are being compared here? Are these the correlations presented in Figure 3? Is this different from the analysis in the previous paragraph?

Line 320: Are there instances where growth loss is not actually reduced growth?

Line 324-325: Does including extreme drought years change the results drastically? I ask because severe drought year frequency is increasing.

Line 327-328: What is drought intensity? How do you define it?

Line 336: if you are talking about a new different analysis, I'd recommend starting a new paragraph

Line 342 paragraph: If this is conducted on detrended data, it is possible that standardization could remove any long term trend associated with changing DTR. I am wondering if this pattern holds if you were to conduct the same analysis, but on data that isn't detrended, but has a tree size or tree age effect in the model.

In lines 342-343, it implies that the moving interval analysis is only done between DTR and growth, but then you say you did this on all climate variables...So was this a joint model approach where you fit mixed effects models?

Lines 347-351: Why use Random forest and linear mixed effects models?

Line 350: So differences in species responses to climate are not included here? Do all species have the same temporal range? Could this impact inferred growth?

Line 353: Linear trends in what? of the DTR-growth loss over time? across space? In response to some other variable?

Figures:

Fig. 1: If space permits, adding a more descriptive figure caption would help the readers interpret this figure.

Also, I have some notes on small improvements to make this figure more visually appealing.

1. Align the text and figures horizontally, so that the "small DTR and large DTR" under the low p50 trees are aligned with that of the high p50 trees.
2. Same thing for the tree growth vs PDSI figures...make sure these are aligned, and the text above them.
3. Make sure the circles are aligned horizontally.
4. If you can, get a image of the thermometer that does not have the checkered grey background.

Fig. 2 The colors on this figure wont all be visible to someone with color blindness. Ideally the color palate should be color-blind friendly. Another suggestion is to color keep the color variations by species, but group colors by similar species...for example all the pines as variations of Blue, and give gymnosperms and angiosperms different color families. This might be challenging to do, but it will help make the dots on the map more meaningful

Fig 3: This figure is hard to read for a variety of reasons. First, the DPI/resolution seems low. Second, all of the violins are very scrunched in, making it hard to really see the distribution shape, and third, the colored dots are all running together.

In the figure caption, there is no a and b panels, just all one. Additionally there are no black dots.

Some suggestions: If comparing dry and wet year correlations within a species is the most important, see if there is a way to make this bigger (split species up until two panels...gymnosperms & angiosperms?). Or, if comparing across species is more important, then split up into dry vs we years. Also, what test are the pvalues and *** for? I think it is for comparing across dry vs wet years but I am not sure based on the caption.

Figure 4. It seems strange that NOPU and QUAL have the exact same drought tolerance?

Figure 6: Where is the relationship between warming rate of Tmax and delta DTR-growth loss correlation? Also it would be good to have a figure between DTR and Tmin/Tmax for all sites/species.

Figure 7: Why go all the way to 1990 when the sample depth drops off rapidly after 1980? What is drought tolerance here? How is it quantified?

Figure S3: Based on this figure and the previous figure—it seems like there is a strong link between DTR correlations and whether the trees are generally responsive to temperature or precipitation. Why is this interaction not explored?

Figure S4: Add error bars to this

Figure S5: Comparison of tree growth indices—which I believe are quantified at the site scale—that it is not absolute growth, so its not super informative to compare across space. This figure is also hard to read—its blurry and the violin plots are very squished together.

Figure S6: DTR is high, but is it not also correlated to what the Tmax is in most cases?

Figure S8. What are each of these panels? Different species, probably? Also it looks like there are not many high DTR sites, or they are just covered by the other dots.

Reviewer #2 (Remarks to the Author):

The manuscript by Zhang et al. is interesting work. I have several questions that I think need to be addressed before I can evaluate how meaningful this contribution is.

Major comments

1. The authors use the ITRDB, which is great resource but there is no discussion about how the

dataset is biased due to only having canopy dominant trees. What kind of impact does this have on the analysis? When thinking about climate's influence of forests, it needs to be clear this is only for canopy dominant trees. Klesse et al. (2018) found that this bias overestimated the impacts of climate change. There needs to be some discussion of this

Klesse, S., DeRose, R. J., Guiterman, C. H., Lynch, A. M., O'Connor, C. D., Shaw, J. D., & Evans, M. E. (2018). Sampling bias overestimates climate change impacts on forest growth in the southwestern United States. *Nature communications*, 9(1), 5336.

2. I understand that the authors detrended the data to remove the biological growth trend from the series. But this really doesn't remove the effect of age. While all the trees in the itrdb are canopy dominant, the ages are quite different. Those ages have been shown to matter using the same itrdb data set in Au et al. 2022. How do you think you could incorporate age? It should at least be somewhat calculable from the itrdb. At the very least worth discussing in the manuscript.

Au, T. F., Maxwell, J. T., Robeson, S. M., Li, J., Siani, S. M., Novick, K. A., ... & Lenoir, J. (2022). Younger trees in the upper canopy are more sensitive but also more resilient to drought. *Nature climate change*, 1-7.

3. Ok, I have a couple of related points. I think the authors first need to present the "non-drought" results, so readers understand how DTR is related to growth. Then present the drought reduction results. Jumping back and forth throughout the manuscript is confusing.

4. Related, the authors used any negative PDSI values as a drought response and any positive value as a wet response. I think those should be cutoff at -0.5 and 0.5 with the middle being normal conditions. I strongly feel the analyses should be redone with this new metric as that is more accurate for what the PDSI represents. What would be more interesting for the paper is to look at how DTR impacts growth for dry, normal, and wet years. Do trees make up losses from drought years during the wet years and is that changing over time with changes in DTR?

5. Building off the last point, On lines 323-325, when calculating growth loss, the authors use PDSI >0.5 to define normal years. This is actually wet! Normal should be -0.5 - 0.5. I suppose the authors could call it non-drought years and include anything > than PDSI=zero but they should remove PDSI >4 in a similar way they do for drought years and the -4 value.

6. The authors found a latitudinal response and a physiological pattern with how DTR impacts growth. The high latitudes are also dominated by conifers. While the lower to mid latitudes are more of a mix but hardwoods are more common. How much of the pattern that the authors find are related to biome the trees live compared to a more physiological difference between conifers and hardwoods? Is it the species drought tolerance, the fact it's a conifer or hardwood, or where the tree is growing that had the largest impact of how DTR impacted growth?

7. Line 150: I feel very strongly that one should not remove outliers without some good description as to why they removed it. I did not see that anywhere in the manuscript and since it appears to have a big influence on the relationship, that should be discusses somewhere.

8. There is a lot of discussion about how photosynthesis and growth are connected. Some recent work suggests these two things are often decoupled, particularly during drought. Perhaps your findings here explain at least one reason that may be the case. Regardless, discussing your findings in the context of that work is really important. Here are a few articles:

Dow, C., Kim, A. Y., D'Orangeville, L., Gonzalez-Akre, E. B., Helcoski, R., Herrmann, V., ... & Anderson-Teixeira, K. J. (2022). Warm springs alter timing but not total growth of temperate deciduous trees. *Nature*, 608(7923), 552-557.

Cabon, A., Kannenberg, S. A., Arain, A., Babst, F., Baldocchi, D., Belmecheri, S., ... & Anderegg, W. R. (2022). Cross-biome synthesis of source versus sink limits to tree growth. *Science*, 376(6594), 758-761.

Kannenber, S. A., Cabon, A., Babst, F., Belmecheri, S., Delpierre, N., Guerrieri, R., ... & Anderegg, W. R. (2022). Drought-induced decoupling between carbon uptake and tree growth impacts forest carbon turnover time. *Agricultural and Forest Meteorology*, 322, 108996.

Anderson-Teixeira, K. J., & Kannenberg, S. A. (2022). What drives forest carbon storage? The ramifications of source-sink decoupling. *The New phytologist*, 236(1), 5-8.

9. Why does the weakening start in 1940? Warming is generally thought to have started in the 1970s and 1980s. Did nighttime warming start in the 1940s?

10. For the moving interval analysis, I think there needs to be a sensitivity test of how the window size changes the results. Variability in correlation values in moving interval analyses can change quite a bit based on the window length. How does changing the window length influence your results?

Minor comments

Line 21: Add the letter a in "with a high night warming rate"

Line 40: Change litter to little. Also, define "between them"

Line 290: When was the SPEI used, did I miss it?

Sincerely,
Justin Maxwell

Reply to reviewer #1's comments

REVIEWER COMMENTS

Reviewer #1 (Remarks to the Author):

Review of Fading regulation of diurnal temperature ranges on drought induced growth loss

Comments for Author

GENERAL COMMENT: *Overall goal: Use ITRDB to evaluate how DTR affects species-level responses to drought (drought-induced growth loss) across space, and time.*

Overall summary:

The authors leverage a large, multi-species database of tree ring time series data to assess the effect of Diurnal temperature range on drought induced growth loss.

Because they looked at several different taxa across regions that are generally controlled by different drivers (temperature vs precipitation), they find that the effects of DTR on relative tree growth vary widely across species, under different drought vs non-drought conditions, and across space. The authors then use several analyses to try to distill these differences into some common findings.

Noteworthy results & Significance: Quantifying the relationships between physiological drought tolerance parameters and observed DTR-growth loss correlation coefficient is a unique, possibly new approach to link observed growth to physiological metrics. The findings of this work have relevance to predicting responses of trees to future changes in DTR, and in understanding past climates.

This manuscript has several strengths, and the authors research questions are both relevant and timely. Specifically, the key strengths and noteworthy results of this manuscript are:

- 1. Leveraging a wide network of tree ring data available should allow the authors to quantify differences in the effects of DTR across species, and say something about how it varies across space.*
- 2. The effects of Diurnal Temperature Ranges on drought response are an important*

topic needed to address how asymmetric changes in high vs. low temperatures could enhance or alleviate negative effects of drought stress.

3. Quantifying the relationships between physiological drought tolerance parameters and observed DTR-growth loss correlation coefficient is a unique, possibly new approach to link observed growth to physiological metrics.

RESPONSE: Thank you very much for your positive evaluation of our manuscript.

However, there are some methodology, organization, and clarity issues that I believe need to be addressed with this manuscript in revision:

GENERAL COMMENT: *1. The analyses focus on DTR and its effect on growth loss during drought, but don't directly quantify the influence of high summer temperatures in driving high DTR, nor does it directly address the relationships between DTR, tmin, and tmax in the dataset. Based on the maps in the supplement, it does seem that the role of DTR likely depends on whether the tree is in a moisture limited or temperature limited system, or at least that DTR, tmin, tmax, and precipitation are linked. I am not totally convinced that the relationships between DTR and growth loss are not a product of VPD or humidity/ snowpack interactions. I wonder if an initial exploration into drivers (or correlations) of DTR with other climate variables could help shape the mechanisms discussed in the discussion.*

Suggestion: Though authors do discuss the role of min vs max temperatures and potential effects of increasing DTR on growing season length, I think that the authors should show the relationships between DTR, tmin, and tmax themselves somewhere in the supplement, as some of the statistical analyses include all three of these variables, which may not be appropriate if they are strongly correlated.

RESPONSE: We thank the reviewer for pointing out that the relationship between DTR, tmin and tmax should be addressed. We have added the correlations between DTR, tmin, and tmax in the supplemental Figure. S10 (Fig. R1 in the response). Maximum/minimum temperature and DTR were highly correlated in most sites. We reworked our statistical analyses (e.g. linear mixed models and structural equation models) to make sure to account for the correlations between

between DTR, tmin, and tmax in the revised manuscript, in lines 340-349.

Fig. R1 Correlations between Maximum temperature (Tmax), minimum temperature (Tmin), DTR, precipitation (PRE) from 1901-1980.

GENERAL COMMENT: 2. The methods section outlines a lot of different statistical analyses, but it is hard to follow what each test is doing, and why you need to report all of these. It is also a little unclear in the methods which values (means, temporal variations, correlation coefficients, individual growth indices or whole chronologies?) are being used and why. This is an issue because it is difficult to evaluate whether the figure/statistical test is necessary, but also because it makes the manuscript hard to follow.

Suggestion: I appreciate that trying to distill all the complexity in the relationship between DTR and growth loss is a heavy lift, but I think some clarity and organization in the methods section details would help—see statistical analysis comments below. Also, I believe that adding in some key sub headers that are common throughout the discussion, results, and methods would greatly improve the clarity—You already have

the start to this in the Results and Discussion, but I would argue for a few more specific sub headers that describe the single question being addressed with each statistical test/analysis, and to use these same ones to organize the methods section. I also wonder whether all these different analyses are needed in this manuscript.

RESPONSE: We thank the reviewer for the suggestion on how to revise the method section. We have added sub-headers for the various sub-sections, and reorganized the sub-sections following your suggestion. The method section was revised to be more concise and clearer by removing Random Forest method combining some analyses into one (lines 318-391).

GENERAL COMMENT: 3. I think there is a fundamental issue with using detrended growth indices in some (not necessarily all) of the analyses in this paper. Detrended growth indices (whether at the individual tree, or site-chronology), are detrended to be relative growth for that site. So, growth losses are also relative to that site. So, after detrending, if two sites both have a growth loss of 0.5, they could have drastically different absolute changes in growth. There is a good deal of discussion of growth loss, and how growth losses vary across high vs low DTR regions, but you can't really compare absolute growth losses from these detrended time series. These comparisons are discussed (briefly) in the context of changes in Carbon uptake, which is not something you can quantify with growth indices. Additionally in most places in the manuscript "tree growth" should be changed to "relative tree growth," to distinguish that this analysis was done on detrended data.

RESPONSE: We thank the reviewer for pointing the issue of detrended growth indices out. We agree with the reviewer's comments that growth losses are relative to that site. We tried to use raw tree-ring data, but the influences of age and size made them hard to be compared between different sites due to the age and size effects. Hence, we changed the 'tree growth' to 'relative tree growth' or 'tree growth indices' according to the reviewer's comments and we also deleted the discussions on in the context of changes in carbon uptake.

GENERAL COMMENT: 4. Issue: The trees in the ITRDB are biased towards highly sensitive sites or individuals, compared to a more ecologically sampled dataset

(Klesse et al 2018). Thus, they are likely biased in their relationship to DTR.

Suggestion: At the very least these biases should be acknowledged in the manuscript and in the methods.

RESPONSE: We agree with the reviewer that the trees in the ITRDB are biased towards highly sensitive sites. We have acknowledged these biases in the method section according to the reviewer's comments (lines 293-296), where we write: *Most ITRDB samples are selected for particular climate sensitivity, suggesting that they are overly sensitive compared to an ecological sampling network, thus any inferred drought sensitivity/drought loss might also be overestimated.*

GENERAL COMMENT: 5. Issue: Temporal analyses go up to the year 1990, even though the sample depth decreases rapidly after 1980, making the last 2-3 time periods used in these analyses potentially biased by the species and # of individuals present in the dataset.

Suggestion: This either should be discussed/addressed in the discussion, or the authors should limit the time series analysis to 1900-1980.

RESPONSE: We agree with the reviewer's comment that the sample depth drops steeply after 1980 and change to focus our analysis for the period 1900-1980 in the revision. We have removed the last 2 time periods which did not affect our conclusions.

GENERAL COMMENT: Relating to issue #2 above, the following are different statistical analyses that I gathered from the methods section, with a few questions about why these statistical methods are used.

- 1. Correlation of annual growth & annual climate variables*
- 2. Linear regression of annual (?) summer DTR and site-level chronologies, in wet and dry years*
- 3. Linear regression of average site DTR and average growth in wet and dry years*
- 4. Boosting regression tree to identify importance of drivers of drought induced growth loss.*
- 5. Demming regression of DTR and function traits*
- 6. Moving interval response analysis of correlation of DTR & growth loss*

7. *Random Forest Model to detect shifts in contribution of climate variables.*
8. *Linear mixed effects models to quantify changing Contribution of climate variables to growth loss.*
9. *Linear trends fitted to changes in DTR-growth loss correlations from moving interval response analysis.*
10. *Correlation between linear trend of DTR growth loss and warming rate*
11. *Calculated temporal changes in the hydraulic traits and climate growth loss correlation are calculated.*

Why are analyses 1-4 not combined into single linear regression that predicts annual growth (or annual growth loss), from annual DTR, temperature, precipitation, site-level DTR, species, and other covariates used in the boosting regression trees with random slopes for wet and dry years or even species?

RESPONSE: We highly appreciate the reviewer for listing the details of our method section. We had revised the method sections to make it more clearly (lines 317-391), where we write the first paragraph as follows:

Factors influencing tree growth. Correlation between site tree-ring chronologies and site climate (i.e., temperature and precipitation) was calculated to detect the dominant climate factor for each site. We also calculated correlations between site summer (June-August) DTR and tree-ring index for each site to evaluate the influence of summer DTR on tree growth during relatively dry (summer mean PDSI<-0.5) and wet (summer mean PDSI>0.5) conditions.

GENERAL COMMENT: *It's also not clear from the methods and results why both a random forest model and a linear mixed effects models were used to quantify changes in contribution of climate variables to growth loss. The linear trends fitted to the DTR-growth loss correlations are later used to compare to the rate of warming, so these two make sense and are presented in the results, but I wonder why you did also relate them to Tmax?*

RESPONSE: Both a random forest model and a linear mixed effects models were used to confirm the results that quantify changes in contribution of climate

variables to growth loss. We agree with the reviewer here, that there is no need for both. It just makes the manuscript unnecessarily complicated. We removed random forest from the manuscript.

We agree with the reviewer's comments that Tmax was not a key variable in this part of the analysis and we removed these correlations from the revised version. We have also added the correlations with DTR, as they are actually important.

GENERAL COMMENT: You also say that you calculated temporal changes in hydraulic traits and climate growth loss correlation, which is maybe referring to the results in figure 7 (?), but it was difficult to tell, since the authors don't provide enough information to reproduce these analyses.

RESPONSE: Yes, the temporal changes in hydraulic traits and climate-growth loss correlation were shown in Figure.7. We clarify in the revised manuscript that hydraulic traits were assumed to be constant, but climate-growth loss changed with time. Therefore, we could calculate the changes in the correlation between hydraulic traits and climate-growth loss correlation over time. The detailed information was added in the revised manuscript to better describe the methods, in lines 382-385, which now state:

“Species hydraulic traits were assumed to be constant over time, while the DTR-growth loss correlation changed. Temporal changes in the hydraulic traits and climate-growth loss correlation were then calculated to investigate changes in the regulations of hydraulic traits on the DTR-growth loss correlation for the period 1901-1980.”

General line by line comments:

COMMENT: Line 26: Use of regulation here does not state the direction...is it consistently negative? Also asymmetric day and night warming does not necessary state the direction either. Are days or nights heating faster consistently?

RESPONSE: We have changed the sentence to “*The alleviation of reduced DTR on drought-induced growth loss was mainly founded for drought resistance*

species.” to show the direction according to the reviewer’s comments, in lines 22-24.

Days or nights were not heating faster consistently, as you can see the DTR had different trends across sites (Fig. R2).

Fig. R2 The trend of DTR from 1901 to 1980 for each site

COMMENT: Line 40: Litter should be little.

RESPONSE: The word was corrected accordingly.

COMMENT: Line 39: I think there are a few other studies on changing DTR effects on other metrics of growth—NPP possibly?

RESPONSE: We have added the following references on changing DTR effects on NPP in the revised version:

- [1]. Li, H., Wu, Y., Liu, S., & Xiao, J. (2021). Regional contributions to interannual variability of net primary production and climatic attributions. *Agricultural and Forest Meteorology*, 303, 108384.
- [2]. Xu, X., Jiang, H., Guan, M., Wang, L., Huang, Y., Jiang, Y., & Wang, A. (2020). Vegetation responses to extreme climatic indices in coastal China from 1986 to 2015. *Science of The Total Environment*, 744, 140784.
- [3]. Xu, L., Meng, P., Tong, X., Zhang, J., Li, J., Wang, X., ... & Liu, P. (2022). Productivity and water use efficiency of *Pinus tabulaeformis* responses to climate change in the temperate monsoon region. *Agricultural and Forest Meteorology*, 327, 109188.

COMMENT: Line 53: Is dryness here a soil property/characteristic? Or a state of soil moisture. At the beginning of this sentence, I thought you were going to talk about soil properties (%sand, soil type, etc.)

RESPONSE: We refer to the characteristic of soil moisture here. We have changed the wording accordingly.

COMMENT: Line 54: I think this contradicts the statements in previous lines—51-52.

RESPONSE: Thank you for pointing it out. We have revised this sentence to “both day and night warming have been shown to reduce tree growth in extremely dry soils” to be consistent with the lines 51-52.

*COMMENT: Line 61: stomatal**

RESPONSE: The word was corrected accordingly.

COMMENT: Line 63: enhance should be enhanced.

RESPONSE: The word was corrected accordingly.

COMMENT: Line 64: This is missing a word somewhere—field studies supported the hypothesis?

RESPONSE: We had added the missed word.

COMMENT: Line 67-69: Does compensatory photosynthesis actually increase carbon gain? It would have to compensate for the nighttime respiration, and then be enhanced/primed to go above and beyond what was lost in respiration.

RESPONSE: That is an interesting question. We agree with the reviewer’s comments that photosynthesis have to compensate for the nighttime respiration, and then be enhanced/primed to go above and beyond what was lost in respiration. According to the references, the compensatory photosynthesis actually increases carbon gain beyond the additional respiratory losses (Balducci et al. 2016, Cheesman et al. 2013).

[1]. Balducci L, Cuny HE, Rathgeber CB, Deslauriers A, Giovannelli A, Rossi S. Compensatory mechanisms mitigate the effect of warming and drought on wood formation. *Plant Cell Environ* 39, 1338-1352 (2016).

[2]. Cheesman AW, Winter K. Elevated night-time temperatures increase growth in seedlings of two tropical pioneer tree species. *New Phytologist* 197, 1185-1192 (2013).

COMMENT: Line 70. You should introduce the % loss of hydraulic conductivity a little

more here. Is this typically a species-wide parameter or measured over time or site conditions?

RESPONSE: We have added an explanation of percent loss of hydraulic conductivity (PLC) in the revised manuscript. It is a species-wide parameter, with some variation with site conditions within the same species, which we cannot quantify and, therefore, include here.

COMMENT: Line 80-83: Are you using Individual tree responses?

OR site level chronologies? It sounds like individuals from this sentence.

RESPONSE: We used site level chronologies by creating a site chronology for each site. We have revised the sentence to clarify it (line 317-319):

“Correlation between site tree-ring chronology and site climate (i.e., temperature and precipitation) was calculated to detect the dominant climate factor for each site.”

COMMENT: lines 83-85: This sentence is good, but I am wondering if you can change this sentence slightly or add a sentence that briefly overviews how you did this—that is outline the different analyses that you used to investigate the relationship between DTR and drought induced growth loss for each species. This would help with following the methods and results, since there are several analyses that involve correlations with DTR. You also don’t mention looking at the weakening DTR temporal effects at all here, but it should be included here as it is in the title of the paper.

RESPONSE: We thank the reviewer for this valuable suggestion. We have added some sentences as a brief overview of the method in investigating the relationship between DTR and drought induced growth loss for each species according to reviewer’s comment (lines 83-90), and we had also added a sentence to show the hypothesis of weakening DTR temporal effects (lines 90-92):

We investigated the relationship between DTR and drought-induced growth loss for each species and obtained species-specific hydraulic traits to test their relevance to explain the interaction between DTR, drought, and growth loss. Temporal changes of DTR effects on drought-induced growth loss from 1901 to 1980 was quantified using linear mixed models. We hypothesized that species’ stem hydraulic vulnerability would positively influence the regulation of DTR on drought-induced tree growth

loss.

COMMENT: Lines 91-95: Most of these correlation distributions are overlapping with zero according to the figure. Could you define the threshold at which you determined these are positive or negative correlations. Also you should say what the significance values are testing in the figure. I think it is differences within species in dry and wet years.

RESPONSE: We have redrawn Fig. 3 according to the reviewer's comments. In the revised manuscript, we use a revised figure to illustrate the DTR-growth correlation during dry and wet years (Fig. R3).

Fig. R3. Partial correlation between DTR and growth considering influences of temperature during dry (a) and wet (b) years for each species.

COMMENT: Line 94: Is it site-level DTR, or annually varying? I think if it is

correlations in dry years then it is annually varying?

RESPONSE: Here we used the site-level summer (June-August) DTR. The figure showed the correlation between site-level summer DTR and tree growth in dry years and wet years.

COMMENT: Line 96: spelling error—should be *Quercus spp.*

RESPONSE: Thanks! We have corrected it.

COMMENT: Also, Perhaps you should explain in the methods why you lump some *quercus* all together in *Quercus spp.*, and separate others. I think it has to do with the structure of the data in the database, but I am not sure.

RESPONSE: Yes, this is due to a structural quirk of the ITRDB. *Quercus spp* are reported separately from individual *Quercus* species. *Quercus spp.* was separated from other *Quercus* species in the ITRDB. We used the same species classification as it is in the ITRDB. We have changed the sentence to: *It should be noted that some oak species were lumped as Quercus spp. (QUSP) in the ITRDB (line 282-283).*

COMMENT: Line 105: you spell out all the scientific names except FASY here.

RESPONSE: The scientific name of FASY has been spelled out prior to this mention (line 95), so we did not spell it out in line 105.

COMMENT: Lines 108-115: You are comparing growth indices, not growth here, so make sure to state that. Because the data is detrended by site, its hard to actually say anything about growth across space.

RESPONSE: We agree with the reviewer's comments. We have changed all the 'tree growth' to 'relative tree growth' or 'growth indices' in the revised manuscript.

COMMENT: Lines 118-119: so this is the mean DTR vs mean drought induced growth loss over time for all chronologies of that species? individuals?

RESPONSE: Yes, it is the site mean DTR vs site mean drought induced growth loss over time for each chronology of that species, not individuals. We conducted this analysis at the species-level, not individual level to detect the influences of DTR on growth across the species' distribution.

COMMENT: Line 122: What about LASI, QUMA? Thes also look negative in fig. S7.

RESPONSE: Yes, DTR had negative influences on growth loss for the two species.

They have been added to the sentence. Thank you for pointing out this oversight.

COMMENT: Lines 131-135: How much variance does p50 or p88 explain in the relationship?

RESPONSE: We are sorry for forgetting to add the explained variance in the relationship. P50 or P88 explained 61% of total variance in the relationship (line 136). Both variables had significant influences on drought-induced growth loss.

COMMENT: Lines 140-142: Why use a random forest model to detect changes, and then a linear effects model? Why not include the temporal component in the regression model?

RESPONSE: We agree with the reviewer here, that there is no need for both. It just makes the manuscript unnecessarily complicated. We have removed Random Forest from the manuscript.

We have added the age effect as the temporal components in the regression model in the revised manuscript accordingly: *In these analyses, DTR, PDSI, Tmax and Tmin were included as fixed effects, while age and species were included as random effects.* (lines 371-372)

COMMENT: Line 147: “relative” should be “relatively”

RESPONSE: The word has been corrected accordingly.

COMMENT: Also the effect of PDSI is the strongest over time, but does also increase according to figure 4. These variables (PDSI and DTR) are likely correlated, along with Tmin and Tmax...how do you parse these effects?

RESPONSE: Yes, these variables are correlated with each other, as we shown in Fig. S10 of the revised manuscript (Fig. R1 in the response). Because DTR was highly correlated with Tmax and Tmin, we used a linear mixed model that used only PDSI, DTR and their interaction as fixed variables, the results showed that the DTR had increasingly effects on growth loss. The separated effects of PDSI and DTR could be parsed based on linear mixed models.

COMMENT: Line 153-154: This is interesting...is it because maximum temperature is also increasing rapidly? Or is there an interplay between VPD/moisture availability and DTR/min temperature that is not looked at?

RESPONSE: From our data, we cannot determine the response, but we have added some text to the revised manuscript (lines 152-155) to reflect the potential influences of changing rate of maximum temperature on the DTR-growth loss correlation. We did see that high warming rates of Tmax and Tmin correspond to negative changes in DTR-growth loss correlation (Fig. R4). We have added the related information: *Temporal changes in DTR-growth loss correlation were negatively influenced by the warming rate of Tmin and Tmax (Fig. S9). Increase in warming rate of both Tmin and Tmax would lead to decrease in DTR-growth loss correlation. Consequently, how the relationship between DTR and drought-induced growth loss changed was influenced by the asymmetric warming rate of between Tmax and Tmin.*

Fig. R4 Relationship between changes in warming rate of Tmax and changes in the DTR-growth loss correlation

COMMENT: Line 176: *This makes sense...lower temps at night mean some relief from high temps*

RESPONSE: We thank the reviewer for the supportive comment.

COMMENT: Line 176-179: *There is quite a lot packed into this sentence. I think there*

should be a comma after tree species on line 177.

RESPONSE: A comma has been added after tree species on line 177 accordingly (line 175).

COMMENT: Line 183: One question that keeps coming up for me while reading this is: how do you parse the effects of high DTR from the effects of high temperatures during drought? They are linked, right? Okay this gets acknowledged around line 189, but DTR is both high daytime temperature and low night time temperatures.

RESPONSE: Yes, the effects of high DTR are linked with the effects of high temperatures during drought. We have calculated partial correlation of DTR on growth during drought to attribute the influence of Tmax and parse the effects of high DTR from the effects of high temperatures during drought (Fig. R5), and the figure was added as Fig.S2 in the revised manuscript. This result confirmed that DTR had negative influence on tree growth during drought.

Fig. R5 Partial correlation between DTR and growth with the influence of Tmax during drought years

COMMENT: Line 193....”what refers to...” I really fit what this sentence is trying to say, I don’t think.

RESPONSE: We have revised the sentence to “*High daytime temperatures may lead to drought-induced tree growth loss due to reduced stomatal conductance and photosynthesis at particularly high temperatures*” to clearly address what we want to say in the revised manuscript accordingly (line 190-192).

COMMENT: Line 199-202: Its not clear how increasing respiration would alleviate drought stress. If photosynthesis is compensating for increased respiration, it is just a net zero, right? Unless nighttime warmer temperatures led to an increase in photosynthesis above and beyond what was spent in respiration, or the normal amount of photosynthesis??

RESPONSE: Yes, we agree with the reviewer's comments. From our results, it is likely that nighttime warmer temperatures lead to an increase in photosynthesis above and beyond what was lost due to the additional nighttime respiration. The mechanism beyond this phenomenon has been explored in related references, which found increased growth with compensatory mechanism as the compensatory photosynthesis actually increases and leads to a net carbon gain (Balducci et al. 2016, Cheesman et al. 2013). We have revised these sentences to address this possible mechanism according to the reviewer's comments: *Our results suggested that warmer nighttime temperatures led to an increase in photosynthesis above and beyond what was spent in respiration (Line 200-201).*

- [1]. Balducci L, Cuny HE, Rathgeber CB, Deslauriers A, Giovannelli A, Rossi S. Compensatory mechanisms mitigate the effect of warming and drought on wood formation. *Plant Cell Environ* 39, 1338-1352 (2016).
- [2]. Cheesman AW, Winter K. Elevated night-time temperatures increase growth in seedlings of two tropical pioneer tree species. *New Phytologist* 197, 1185-1192 (2013).

COMMENT: Lines 201: Is it also possible that the wider rings stem from warmer nights, and increased snowpack melt in areas where growth is limited by snowpack?

RESPONSE: Yes, we agree with you that it is possible. We thank you for providing this possible explanation, and we have added it in the revised manuscript: *It is also possible that wider rings stem from warmer nights and increased snowmelt in areas where growth is limited by snowpack and/or growth limited by cooler night time temps, such as in the northern boreal forests (line 204-206).*

COMMENT: Lines 207-208: High DTR could also be related to available moisture/humidity. For exemplar drier desert areas have larger DTR's, but because of their low humidity.

RESPONSE: We agree with your comments. In fact, we observed a negative

correlation between DTR and precipitation (Fig. R6), which we present as Fig.S3b in the revised manuscript. We have also added the following text to the revised manuscript: *High DTR could be related to available moisture/humidity* (line 176-177).

Fig. R6 Correlation between DTR and precipitation for each sample site.

COMMENT: Is it compensation, or is growth limited by cooler night time temps in some places? Or more likely, since you are looking at yearly variables...its a longer growing season as stated in 207-210.

RESPONSE: We thank the reviewer for pointing out this possible additional mechanism. We think that all three (compensation, cold nights and longer growing season) are possible mechanisms that may explain why warmer nights alleviate drought stress in regions with low DTR. We have revised the section as follows: “*It is also possible that wider rings stem from warmer nights and increased snowmelt in areas where growth is limited by snowpack and/or growth limited by cooler night time temps, such as in the northern boreal forests. Furthermore, the leaf-on period and potentially the growing season length may be also affected by asymmetric warming*” (line 204-208).

COMMENT: 212-213: this sentence could be stronger if you say the direction of the influence of DTR.

RESPONSE: We have revised the sentence to ‘*The alleviation of reduced DTR on drought-induced growth loss was stronger for drought-tolerant species*’ to show the direction of the influence of DTR (line 218-219).

COMMENT: line 216, and this occurs elsewhere. The use of past tense to describe results of previous studies, but without distinguishing it from the results of this study in some way. For this sentence you could say high hydraulic safety generally corresponds to lower stomatal conductance (cite), following a well established tradeoff...

RESPONSE: We thank you for pointing out this issue. We have revised these sentences and carefully checked the discussion section to avoid similar issues.

COMMENT: Line 218-219: But is this just because of really high temps?

RESPONSE: High temperatures with low precipitation lead to exacerbated drought conditions. We have revised the sentence to “*High DTR can exacerbate drought conditions and has been shown to lead to low stomatal conductance during droughts and reduced growth eventually.*” in lines 224-226 to show the link between high DTR and exacerbated drought.

COMMENT: Line 223: This section title is confusing. Is the hydraulic trait regulation weakened? Or is it the relationship between DTR and growth loss.

RESPONSE: It is the hydraulic trait regulation that is weakened over time. We have revised the section title to ‘*Decoupling of relationship between hydraulic traits and DTR-growth loss relationship*’ (line 229-230).

COMMENT: Line 269-286: Another issue with the database to highlight is that most ITRDB samples are selected for particular climate sensitivity, suggesting that they are overly sensitive compared to an ecological sampling network (Klesse et al 2018.). This isn’t a dealbreaker for this study, but you should point out that any inferred drought sensitivity/drought loss might also be overestimated here.

RESPONSE: We thank you for pointing out this issue. We now do point out that any inferred drought sensitivity/drought loss might be overestimated due to the high climate sensitivity of the ITRDB in our revised manuscript accordingly: *Most ITRDB samples are selected for particular climate sensitivity, suggesting that they are overly sensitive compared to an ecological sampling network, thus any inferred*

drought sensitivity/drought loss might also be overestimated (lines 293-296).

COMMENT: Line 295: This sentence is a little confusing for me.

RESPONSE: We want to express that lethal water potential represents a species' drought tolerance. Low lethal water potential indicates high drought resistance, while high lethal water potential indicates low drought resistance. We have revised the sentence to "*Lethal water potential represents a species' drought tolerance.*" to avoid confusion (line 307-308).

COMMENT: Line 297: define psi 88 too. Its implied but good to define it

RESPONSE: We have added the definition of p88 (xylem pressure at which 88% of conductivity is lost) accordingly (line 309-310).

COMMENT: Line 299: Define hydraulic safety margin too.

RESPONSE: We have added the definition of hydraulic safety margin (defined as differences between naturally occurring xylem pressures and pressures that would cause hydraulic dysfunction) in the revised manuscript (line 312-313).

COMMENT: Line 306: Why Summer DTR? How is summer defined across all these sites?

RESPONSE: Because DTR had the strongest influences on tree growth in summer months (i.e., June-August). Summer climate variables were also used to detect drought-induced growth loss in Au et al. 2022. We have defined summer as June-August and added the following sentence: *We also calculated correlations between site summer (June-August) DTR and tree-ring index for each site to evaluate the influence of summer DTR on tree growth during relatively dry (summer mean PDSI<-0.5) and wet (summer mean PDSI>0.5) conditions (line 319-322).*

Au, T. F., Maxwell, J. T., Robeson, S. M., Li, J., Siani, S. M., Novick, K. A., ... & Lenoir, J. (2022). Younger trees in the upper canopy are more sensitive but also more resilient to drought. *Nature climate change*, 1-7.

COMMENT: Line 309: So there is site-level DTR and a site summer DTR?

RESPONSE: Yes, we used summer mean of DTR at the site-level.

COMMENT: Line 310-311: Wondering why the averages are being compared here...

Wondering why the averages are being compared here? Are these the correlations presented in Figure 3? Is this different from the analysis in the previous paragraph?

RESPONSE: We aimed to compare the relative tree growth between regions with different DTR during dry years and wet years. The averages were being compared to show how relative tree growth changes in regions with different DTR. These correlations are indicated in Fig. S6. This is different from the analysis in the previous paragraph as this paragraph focus on the tree relative growth between different DTR regions, while the previous paragraph focused on how DTR influenced tree growth in each site.

COMMENT: Line 320: Are there instances where growth loss is not actually reduced growth?

RESPONSE: Yes, if there is a negative growth loss, it is actually an enhanced growth. Only when the growth loss is positive, it means reduced growth.

COMMENT: Line 324-325: Does including extreme drought years change the results drastically? I ask because severe drought year frequency is increasing.

RESPONSE: Including extreme drought years would not change the results. They are only very few years that having the PDSI below -4. But it is difficult to explain the mechanism behind this phenomenon because reduced DTR only alleviated tree growth under non-severe drought conditions from previous studies (e.g. Zhang et al. 2022). Hence, we have excluded the extreme drought years here.

Zhang X, et al. Reduced diurnal temperature range mitigates drought impacts on larch tree growth in North China. Science of The Total Environment 848, (2022).

COMMENT: Line 327-328: What is drought intensity? How do you define it?

RESPONSE: We had redefined the drought-induced growth loss according to Au et al. 2022, and the term drought intensity was removed in the revised manuscript. Au TF, et al. Younger trees in the upper canopy are more sensitive but also more resilient to drought. Nature Climate Change 12, 1168-1174 (2022).

COMMENT: Line 336: if you are talking about a new different analysis, I'd recommend starting a new paragraph

RESPONSE: We have split this into two paragraphs in the revised version.

COMMENT: Line 342 paragraph: If this is conducted on detrended data, it is possible that standardization could remove any long term trend associated with changing DTR.

I am wondering if this pattern holds if you were to conduct the same analysis, but on data that isn't detrended, but has a tree size or tree age effect in the model.

RESPONSE: We re-did the analysis using a linear mixed model with PDSI and DTR as fix effects, and species and age as random effects based on raw growth data (Table R1). The results are similar. DTR has equally high contributions to drought-induced growth loss with temporal changes.

Table R1 Temporal change of variable importance identified using linear mixed models (GL~PDSI+DTR+DTR*PDSI+(1|species/age))

Periods	1901-1940	1905-1945	1910-1950	1915-1955	1920-1960	1925-1965	1930-1970	1935-1975	1940-1980
PDSI	-0.05	0.08	0.11	0.18	0.25	0.22	0.15	0.11	0.32
DTR	0.19	0.16	0.19	0.10	0.12	0.19	0.19	0.19	0.17
DTR*PDSI	0.04	-0.09	-0.15	-0.26	-0.32	-0.31	-0.22	-0.22	-0.43

COMMENT: In lines 342-343, it implies that the moving interval analysis is only done between DTR and growth, but then you say you did this on all climate variables...So was this a joint model approach where you fit mixed effects models?

RESPONSE: The moving interval analysis was done for addressing the relationship between DTR and growth. We first detected the changes in the correlation between DTR and growth using correlation analysis. To further addressed the interactions of DTR and PDSI on growth loss, we used linear mixed model to detect the effects of DTR, PDSI, and their interactions on growth loss with DTR and PDSI as climate variables, instead of all the climate variables. We have changed the description of the sentence in question to make it clear that not all the climate variables were used here, with the revised sentence as: *In these analyses, DTR, PDSI, Tmax and Tmin were included as fixed effects, while age and species were included as random effects (line 371-372).*

COMMENT: Lines 347-351: Why use Random forest and linear mixed effects models?

RESPONSE: We agree with you that there is no need for both. It just makes the manuscript unnecessarily complicated. We have removed Random Forest from the manuscript. .

COMMENT: Like 350: So differences in species responses to climate are not included here? Do all species have the same temporal range? Could this impact inferred growth?

RESPONSE: In this analysis, we aimed to quantify the changing contribution of DTR to growth loss using linear mixed model for each species. Species response to climate was highly correlated to the corresponding climate variable, but species-specific responses to climate were not included in these models.

All the species had the same temporal range. We had used linear mixed model and Random Forest to detect temporal changes in variable importance, while linear mixed models obtained similar results with Random Forest. Therefore, this did not impact our results.

COMMENT: Line 353: Linear trends in what? of the DTR-growth loss over time? across space? In response to some other variable?

RESPONSE: Linear trends of the DTR-growth loss correlation over time for each species. This analysis was conducted to show how DTR-growth loss correlation changes over time for each species. Correlation coefficients between the linear trend of DTR-growth loss correlation and warming rate of maximum and minimum temperatures were calculated to detect the influence of changing maximum and minimum temperatures on the changes of DTR-growth loss relationship. The sentence has been revised as: *Linear trends of changing DTR-growth loss correlations over time were calculated for each species (line 376-377).*

Figures:

COMMENT: Fig. 1: If space permits, adding a more descriptive figure caption would help the readers interpret this figure.

Also, I have some notes on small improvements to make this figure more visually appealing.

- 1. Align the text and figures horizontally, so that the “small DTR and large DTR” under the low p50 trees are aligned with that of the high p50 trees.*
- 2. Same thing for the tree growth vs PDSI figures...make sure these are aligned, and the text above them.*
- 3. Make sure the circles are aligned horizontally.*
- 4. If you can, get an image of the thermometer that does not have the checkered grey*

Editorial Note: Elements of Fig. R7 have been redacted as indicated to remove third-party material where no permission to publish could be obtained.

background.

RESPONSE: We thank you for providing detailed suggestions to modify the figure.

The figure and its caption have been revised and are now much improved (Fig.R7).

Fig.R7 Revised Figure.1 A schematic diagram of how DTR regulated drought-induced growth loss.

COMMENT: Fig. 2 The colors on this figure wont all be visible to someone with color blindness. Ideally the color palate should be color-blind friendly. Another suggestion is to color keep the color variations by species, but group colors by similar species...for example all the pines as variations of Blue, and give gymnosperms and angiosperms different color families. This might be challenging to do, but it will help make the dots on the map more meaningful

RESPONSE: The colors in the figure were revised according to the reviewer's comments (Fig. R8). Gymnosperms were given blue colors and angiosperms were given red colors.

Fig. R8 Revised Figure 2

COMMENT: Fig 3: This figure is hard to read for a variety of reasons. First, the DPI/resolution seems low. Second, all of the violins are very scrunched in, making it hard to really see the distribution shape, and third, the colored dots are all running together.

In the figure caption, there is no a and b panels, just all one. Additionally there are no black dots.

*Some suggestions: If comparing dry and wet year correlations within a species is the most important, see if there is a way to make this bigger (split species up until two panels...gymnosperms & angiosperms?). Or, if comparing across species is more important, then split up into dry vs we years. Also, what test are the pvalues and *** for? I think it is for comparing across dry vs wet years but I am not sure based on the caption.*

RESPONSE: The figure was revised according to your comments (Fig. R9).

Fig. R9 Revised Fig.3: Correlation of DTR with tree growth for each species during dry years (a) and wet years (b).

COMMENT: Figure 4. It seems strange that NOPU and QUAL have the exact same drought tolerance?

RESPONSE: We thank the reviewer for pointing out this mistake. We used p50 for NOPU and p88 for QUAL in the previous manuscript. We should use p88 for NOPU as it is a broadleaf species. However, we had not found p88 for NOPU in the references.

COMMENT: Figure 6: Where is the relationship between warming rate of Tmax and delta DTR-growth loss correlation? Also it would be good to have a figure between DTR and Tmin/Tmax for all sites/species.

RESPONSE: The relationship between warming rate of Tmax and delta DTR-growth loss correlation has been added in Fig. R10.

Fig. R10 Relationship between changes in warming rate of Tmax/Tmin and changes in the DTR-growth loss correlation

COMMENT: Figure 7: Why go all the way to 1990 when the sample depth drops off rapidly after 1980?

What is drought tolerance here? How is it quantified?

RESPONSE: We had removed the period after 1980 after considering the drop-off in sample depth, which did not influence our conclusion. Drought tolerance was quantified as P50 for gymnosperm and P88 for angiosperm. We have added its information to the revised manuscript.

COMMENT: Figure S3: Based on this figure and the previous figure—it seems like there is a strong link between DTR correlations and whether the trees are generally responsive to temperature or precipitation. Why is this interaction not explored?

RESPONSE: There is a strong link between DTR correlation and whether the trees are generally responsive to temperature or precipitation (Fig. R11). We have added description of this relationship in the revised manuscript as: *DTR-growth*

correlations were positive when growth was primarily related to temperature, but negative when precipitation had a stronger correlation with growth during dry years (Fig. S4) (line 107-109).

Fig. R11 Relationship between DTR correlations and trees' response to temperature (a) or precipitation (b) during dry years.

COMMENT: *Figure S4: Add error bars to this*

RESPONSE: **We have removed this figure after revising the text.**

COMMENT: *Figure S5: Comparison of tree growth indices—which I believe are*

quantified at the site scale—that is it is not absolute growth, so its not super informative to compare across space. This figure is also hard to read—its blurry and the violin plots are very squished together.

RESPONSE: We have redrawn this figure (Fig. R12) in the revised manuscript.

Fig. R12 Comparison of tree growth between regions with high and low DTR small regions during non-drought (a) and drought years (b) for each species.

COMMENT: Figure S6: DTR is high, but is it not also correlated to what the Tmax is in most cases?

RESPONSE: DTR is highly correlated with Tmax in most cases (Fig. R13). However, we aimed to quantify the relative contribution of Tmax and DTR on growth loss, so we calculated the results for the two variables.

Fig. R13 Correlation between DTR and Tmax for each site

COMMENT: Figure S8. What are each of these panels? Different species, probably? Also it looks like there are not many high DTR sites, or they are just covered by the other dots.

RESPONSE: We have removed this figure as the reviewer commented that the growth was only relative growth.

Reply to reviewer #2's comments

Reviewer #2 (Remarks to the Author):

GENERAL COMMENT: The manuscript by Zhang et al. is interesting work. I have several questions that I think need to be addressed before I can evaluate how meaningful this contribution is.

RESPONSE: We thank the reviewer for the supportive comments and for providing many valuable comments that helped us improve our manuscript. We have thoroughly revised the manuscript according to the reviewer's comments and provide detailed answers below.

Major comments

GENERAL COMMENT: 1. The authors use the ITRDB, which is great resource but there is no discussion about how the dataset is biased due to only having canopy dominant trees. What kind of impact does this have on the analysis? When thinking about climate's influence of forests, it needs to be clear this is only for canopy dominant trees. Klesse et al. (2018) found that this bias overestimated the impacts of climate change. There needs to be some discussion of this

Klesse, S., DeRose, R. J., Guiterman, C. H., Lynch, A. M., O'Connor, C. D., Shaw, J. D., & Evans, M. E. (2018). Sampling bias overestimates climate change impacts on forest growth in the southwestern United States. Nature communications, 9(1), 5336.

RESPONSE: We thank the reviewer for pointing out this issue with the dataset. We have added the discussion about potential biases in the method section, highlighting that the bias may lead to an overestimation of the effects of climate (line 293-296): *Most ITRDB samples are selected for particular climate sensitivity, suggesting that they are overly sensitive compared to an ecological sampling network, thus any inferred drought sensitivity/drought loss might also be overestimated. The related references were also added in the manuscript.*

GENERAL COMMENT: 2. I understand that the authors detrended the data to remove the biological growth trend from the series. But this really doesn't remove the effect of age. While all the trees in the itrdb are canopy dominant, the ages are quite

different. Those ages have been shown to matter using the same itrdb data set in Au et al. 2022. How do you think you could incorporate age? It should at least be somewhat calculable from the itrdb. At the very least worth discussing in the manuscript.

Au, T. F., Maxwell, J. T., Robeson, S. M., Li, J., Siani, S. M., Novick, K. A., ... & Lenoir, J. (2022). Younger trees in the upper canopy are more sensitive but also more resilient to drought. Nature climate change, 1-7.

RESPONSE: We agree with you that the age should be considered. Because our analysis was conducted based on site chronologies, we cannot separate our samples into different age groups like Au et al. (2022) did. However, we have removed chronologies with young average age (<140 years) in the revised manuscript, as Au et al. 2022 point out that these younger trees were particularly drought sensitive. Because all the trees in the same site were cross-dated, there are high correlation between chronologies developed with all trees and chronologies developed with those removing young age trees (Fig. R14, ring width data was used ak006 in ITRDB). Therefore, long chronologies including young trees would not influence our results. The revision had been added in the method section as: *once we removed all sites with no tree above 140 years of age, as young trees were more sensitive to droughts (line 280-281).*

Fig. R14 Chronology developed with all trees and those developed with those removing young age trees (ak006).

GENERAL COMMENT: 3. *Ok, I have a couple of related points. I think the authors first need to present the “non-drought” results, so readers understand how DTR is related to growth. Then present the drought reduction results. Jumping back and forth*

throughout the manuscript is confusing.

RESPONSE: We thank the reviewer for providing a way to better structure and present our results. We revised the results accordingly (line 96-122).

GENERAL COMMENT: 4. Related, the authors used any negative PDSI values as a drought response and any positive value as a wet response. I think those should be cutoff at -0.5 and 0.5 with the middle being normal conditions. I strongly feel the analyses should be redone with this new metric as that is more accurate for what the PDSI represents. What would be more interesting for the paper is to look at how DTR impacts growth for dry, normal, and wet years. Do trees make up losses from drought years during the wet years and is that changing over time with changes in DTR?

RESPONSE: We agree with you that how DTR impacts growth should be calculated for dry, normal, and wet years. We redid the analysis for dry, normal, and wet years with the suggested cut-offs. The growth reduction was calculated by comparing the drought year growth and non-drought year growth as defined in Au et al. 2022. Our results showed that the trees make up growth loss from dry years, which is changing over time with changes in DTR.

GENERAL COMMENT: 5. Building off the last point, On lines 323-325, when calculating growth loss, the authors use $PDSI > 0.5$ to define normal years. This is actually wet! Normal should be $-0.5 - 0.5$. I suppose the authors could call it non-drought years and include anything $>$ than $PDSI = \text{zero}$ but they should remove $PDSI > 4$ in a similar way they do for drought years and the -4 value.

RESPONSE: We thank the reviewer for pointing out this inconsistency and providing a way to better address the results. We redid the analysis according to the reviewer's comments. We had revised the related sentences as: $\overline{Growth}_{non-drought}$ the mean tree growth indices in non-drought years ($4 > PDSI > 0.5$), and $\overline{Growth}_{drought}$ is the mean tree growth indices in years with mild to severe drought ($-4 < PDSI < -0.5$) (line 333-335).

GENERAL COMMENT: 6. The authors found a latitudinal response and a physiological pattern with how DTR impacts growth. The high latitudes are also

dominated by conifers. While the lower to mid latitudes are more of a mix but hardwoods are more common. How much of the pattern that the authors find are related to biome the trees live compared to a more physiological difference between conifers and hardwoods? Is it the species drought tolerance, the fact it's a conifer or hardwood, or where the tree is growing that had the largest impact of how DTR impacted growth?

RESPONSE: We agree with you that the high latitudes are dominated by conifers while the lower to mid latitudes are more of a mix but hardwoods are more common. We tested whether there is a difference in the impact of how DTR impacted growth between the conifer (PCAB) and hardwood (QURO and FASY) in the same temperate biome (Fig. R15). Our results showed that the DTR-growth loss was mainly related to their drought tolerances (Fig. R16). This analysis seems to indicate that it is the species' drought tolerance that had the largest impact of how DTR regulated growth, rather the biome the trees live.

Fig. R15 Distribution of three species in the same (temperate) biome

Fig. R16 Relationship between drought tolerance and DTR-growth loss correlation for the above three species

GENERAL COMMENT: 7. Line 150: *I feel very strongly that one should not remove outliers without some good description as to why they removed it. I did not see that anywhere in the manuscript and since it appears to have a big influence on the relationship, that should be discusses somewhere.*

RESPONSE: **This was an oversight on our side and we completely agree with you that removed outliers need to be justified appropriately. We have redrawn the figure with all the data for the revised version instead of removing the data (Fig. R17).**

Fig. R17 Relationship between changes in warming rate of Tmax/Tmin and changes in the DTR-growth loss correlation

GENERAL COMMENT: 8. *There is a lot of discussion about how photosynthesis and growth are connected. Some recent work suggests these two things. Perhaps your findings here explain at least one reason that may be the case. Regardless, discussing your findings in the context of that work is really important. Here are a few articles:*

Dow, C., Kim, A. Y., D'Orangeville, L., Gonzalez-Akre, E. B., Helcoski, R., Herrmann, V., ... & Anderson-Teixeira, K. J. (2022). Warm springs alter timing but not total growth of temperate deciduous trees. Nature, 608(7923), 552-557.

Cabon, A., Kannenberg, S. A., Arain, A., Babst, F., Baldocchi, D., Belmecheri, S., ... & Anderegg, W. R. (2022). Cross-biome synthesis of source versus sink limits to tree growth. Science, 376(6594), 758-761.

Kannenberg, S. A., Cabon, A., Babst, F., Belmecheri, S., Delpierre, N., Guerrieri, R., ... & Anderegg, W. R. (2022). Drought-induced decoupling between carbon uptake and tree growth impacts forest carbon turnover time. Agricultural and Forest Meteorology, 322, 108996.

Anderson-Teixeira, K. J., & Kannenberg, S. A. (2022). What drives forest carbon

storage? *The ramifications of source-sink decoupling. The New phytologist*, 236(1), 5-8.

RESPONSE: We thank the reviewer for providing so many valuable papers on the relationship between photosynthesis and growth. We have added these to the discussion in the revised manuscript, where we write: *However, photosynthesis and growth are often decoupled, particularly during drought. (lines 214-215)*

GENERAL COMMENT: 9. Why does the weakening start in 1940? Warming is generally thought to have started in the 1970s and 1980s. Did nighttime warming start in the 1940s?

RESPONSE: This particular threshold may be related to the window size (40 years). We agree with you that the warming generally started in the 1970s and 1980s, but warming started in different years in different sites (Fig. R18). Nighttime warming started in the 1940s for some sites, but certainly not all the sites (Fig. R18). This relationship was more dependent on the window size that used to detect the temporal changes. We used 40-year window size, so the weakening appears to have started in 1940. We do not report the exact threshold, as it is dependent on the window size. However, independent of the window size there is a weakening over time.

Fig. R18 warming rate of Tmax and Tmin in the period 1901-1940 and 1941-

GENERAL COMMENT: 10. For the moving interval analysis, I think there needs to be a sensitivity test of how the window size changes the results. Variability in correlation values in moving interval analyses can change quite a bit based on the window length. How does changing the window length influence your results?

RESPONSE: We thank the reviewer for this suggestion. We added a sensitivity test of how the window size changes the results. We tested the window sizes of 30, 35, 40, 45 and 50 years. Our results showed that changes in the window size did not change our conclusion that the effect of DTR on growth loss faded (Fig. R19). However, the exact point when we saw the relationship change changed with window size (see previous comment and response). The initial periods had high correlations of DTR and growth loss for all the window sizes and this relationship faded for all the window size (Fig. R19).

Fig. R19 The sensitive test of window size. P1 to p11 represent the first period to the last period

Minor comments

COMMENT: Line 21: Add the letter a in “with a high night warming rate”

RESPONSE: The sentence had removed in the revise manuscript.

COMMENT: Line 40: Change litter to little. Also, define “between them”

RESPONSE: The sentence has been changed to “*However, whether this relationship may hold true across tree species is still unknown*” to make it more clearly (line 42).

COMMENT: Line 290: When was the SPEI used, did I miss it?

RESPONSE: We are sorry to forget to remove it from a previous version. We have tested the results with SPEI in the place of PDSI in a previous version. They had similar results, so we only report PDSI in the study. The sentence has been removed in the revised manuscript.

REVIEWERS' COMMENTS

Reviewer #1 (Remarks to the Author):

Overall summary (most of this is from my first review): The authors leverage a large, multi-species database of tree ring time series data to assess the effect of Diurnal temperature range on drought induced growth loss. Because they looked at several different taxa across regions that are generally controlled by different drivers (temperature vs precipitation), they find that the effects of DTR on relative tree growth vary widely across species, under different drought vs non-drought conditions, across space, and over time. The authors also explore the effects of changing T_{min} and T_{max} on DTR

Noteworthy results & Significance:

Quantifying the relationships between physiological drought tolerance parameters and observed DTR-growth loss correlation coefficient is a unique, possibly new approach to link observed growth to physiological metrics. The findings of this work have relevance to predicting responses of trees to future changes in DTR, and in understanding past climates.

This manuscript has several strengths, and the authors research questions are both relevant and timely. Specifically, the key strengths and noteworthy results of this manuscript are:

1. Leveraging a wide network of tree ring data available should allow the authors to quantify differences in the effects of DTR across species, and say something about how it varies across space.
2. The effects of Diurnal Temperature Ranges on drought response are an important topic needed to address how asymmetric changes in high vs. low temperatures could enhance or alleviate negative effects of drought stress.
3. Quantifying the relationships between physiological drought tolerance parameters and observed DTR-growth loss correlation coefficient is a unique, possibly new approach to link observed growth to physiological metrics.

In the revised manuscript and comments, the authors put a tremendous amount of work in to address the initial issues with the analyses & organization in this manuscript. Specifically, they added in several new analyses that look at t_{min} and t_{max}, as well as DTR. They also did a good job reorganizing the methods and results section, and clarifying language around growth and growth indices, which I think has vastly improved the flow and specificity of the manuscript. They also addressed all of my questions and comments on sample depth, applicability/biases of ITRDB data, and the analyses in the comments and the manuscript. It's clear that the authors have put a lot of work into the current manuscript, which is clearly written, leverages a wide network of data, and addresses the impacts of important aspect of global climate change (changing DTR) on drought stress.

Minor comments & Issues

Lines 83-84: Minor point most of the trends removed through detrending are likely due to age or site, but they could be disturbances/growth releases, or other processes that drive long term trends. You could just say after detrending temporal growth trends, due to non-climate effects (i.e. age, size, etc).

Lines 96 & Fig. 2: I am still not convinced of the positive influence of DTR on tree growth by this figure alone...it doesnt show which species are significant Could you add asterisks to this to show significance? Or maybe a dotted line to show where 0 is

Lines 107 & Fig S4: These figures are really interesting! It supports the message that water limited systems might suffer under increases in DTR but energy limited systems will benefit from increased DTR.

Lines 120-122: From figure S5, it seems like most of these differences are not significant, so you should qualify this statement.

Lines 129-131: Is figure 7 a simple correlation or output from a random forest model. It looks like a correlation, but the text implies this is from the RF model mentioned above. It would be good to clarify

Lines 136 & Fig 4: What are the colors on figure 4? Just different species? It would be good to have a color bar/legend or color by gymnosperm/angiosperm

Lines 144 & section: I am all in favor of bold titles that help the reader get the main point, but somewhere in this section, I think you need to remind readers that this is assuming that species hydraulic traits are constant over time. You included it in the methods, which is helpful, but you should state that "assuming hydraulic traits for species are relatively constant within the same

trees/population over time..." there is a weakening effect of hydraulic trains on DTR-growth loss relationship. This is a fair assumption and its also what we have data for, but it's also possible that there is variation around p50 p88 values for different populations of species, and that these values could change over time (though, probably not the timescale of interest here).

Fig S8 is compelling, but the all white circles are hard to see and imply that no correlation was done. Can you add a black outline to better visualize the size of the correlation?

Line 149 Become increasingly what?

Line 156 Influences? Instead of influence

Line 169 DTR exerts* should this be plural here?

Line 203: Mitigates should probably be mitigated here?

Paragraph starting at line 188: I appreciate the wide discussion of possible causes in this paragraph! It gives the discussion a lot of context.

Lines 208-209 & throughout: Minor comment: There are a few places where past tense is used to describe results found in other studies, but unless this was shown with this data or in this study I think it needs a qualifier or description of where it was found. For example here, I'd change it to "was enhanced in other studies" or some qualifier.

Line 258: Should this be "have"

Line 265-266: Could you also say that physiological site/tree level responses may differ from inferred community level responses?

Line 292: And north American trees too?

Line 314: obtained*

Line 317: I think this should be "correlations....were calculated", because it was multiple correlations for different species?

Line 327: Won't the mean tree ring index be close to 1 for all chronologies?

Line 335-337: I know this was a decision that was probably made a long time ago, and I think you have enough in this manuscript, so I dont think you should change it, but it would be *interesting* to see how the severe drought years relate to DTR. Maybe there is another manuscript in the works that digs into this?

Line 353: Random selection with replacement or no replacement?

Fig 6: This version looks great! Could you just add text explaining what the grey and white parts represent on the map

Note on figs S2-S3 & S8 the white colors dont show up on the white background—could you make these a different color? Light grey? Or could you make the country backgrounds a different color? For figure S8, you could add a grey outline to the circles so you can still see the size of the white circles.

Reviewer #2 (Remarks to the Author):

I think the authors did a commendable job addressing both mine and the other reviewer's comments. I do not have any additional concerns and recommend publishing this manuscript.

Reply to reviewer #1's comments

REVIEWERS' COMMENTS

Reviewer #1 (Remarks to the Author):

GENERAL COMMENT: Overall summary (most of this is from my first review): The authors leverage a large, multi-species database of tree ring time series data to assess the effect of Diurnal temperature range on drought induced growth loss. Because they looked at several different taxa across regions that are generally controlled by different drivers (temperature vs precipitation), they find that the effects of DTR on relative tree growth vary widely across species, under different drought vs non-drought conditions, across space, and over time. The authors also explore the effects of changing T_{min} and T_{max} on DTR

Noteworthy results & Significance:

Quantifying the relationships between physiological drought tolerance parameters and observed DTR-growth loss correlation coefficient is a unique, possibly new approach to link observed growth to physiological metrics. The findings of this work have relevance to predicting responses of trees to future changes in DTR, and in understanding past climates.

This manuscript has several strengths, and the authors research questions are both relevant and timely. Specifically, the key strengths and noteworthy results of this manuscript are:

1. Leveraging a wide network of tree ring data available should allow the authors to quantify differences in the effects of DTR across species, and say something about how it varies across space.
2. The effects of Diurnal Temperature Ranges on drought response are an important topic needed to address how asymmetric changes in high vs. low temperatures could enhance or alleviate negative effects of drought stress.
3. Quantifying the relationships between physiological drought tolerance parameters and observed DTR-growth loss correlation coefficient is a unique, possibly new approach to link observed growth to physiological metrics.

In the revised manuscript and comments, the authors put a tremendous amount of work in to address the initial issues with the analyses & organization in this manuscript. Specifically, they added in several new analyses that look at t_{min} and t_{max}, as well as DTR. They also did a good job reorganizing the methods and results section, and clarifying language around growth and growth indices, which I think has vastly improved the flow and specificity of the manuscript. They also addressed all of my questions and comments on sample depth, applicability/biases of ITRDB data, and the analyses in the comments and the manuscript. It's clear that the authors have put a lot of work into the current manuscript, which is clearly written, leverages a wide

network of data, and addresses the impacts of important aspect of global climate change (changing DTR) on drought stress.

RESPONSE: We thank the reviewer for the supportive comments and for providing many valuable comments that helped us improve our manuscript.

Minor comments & Issues

Lines 83-84: Minor point most of the trends removed through detrending are likely due to age or site, but they could be disturbances/growth releases, or other processes that drive long term trends. You could just say after detrending temporal growth trends, due to non-climate effects (i.e. age, size, etc).

RESPONSE: We thank the reviewer for pointing out this improper expression. We had revised the sentence to ‘after detrending temporal growth trends, due to non-climate effects (i.e. age, size, etc).’

Lines 96 & Fig. 2: I am still not convinced of the positive influence of DTR on tree growth by this figure alone...it doesn’t show which species are significant. Could you add asterisks to this to show significance? Or maybe a dotted line to show where 0 is

RESPONSE: We did not compare the differences between different species. We cannot add asterisks to this to show significance because of no comparison was conducted. Instead of that, we have added a dotted line to show where 0 is in Fig.2 accordingly.

Lines 107 & Fig S4: These figures are really interesting! It supports the message that water limited systems might suffer under increases in DTR but energy limited systems will benefit from increased DTR.

RESPONSE: Yes, water limited systems might suffer under increases in DTR but energy limited systems will benefit from increased DTR.

Lines 120-122: From figure S5, it seems like most of these differences are not significant, so you should qualify this statement.

RESPONSE: We have qualified this statement by adding following sentence: ‘DTR showed non-significant influences on tree growth for more than 10 species during both dry and non-drought years.’

Lines 129-131: Is figure S7 a simple correlation or output from a random forest model. It looks like a correlation, but the text implies this is from the RF model mentioned above. It would be good to clarify

RESPONSE: Figure S7 is a simple correlation. We had clarified this by adding ‘correlation analysis showed that’ before the sentence.

Lines 136 & Fig 4: What are the colors on figure 4? Just different species? It would be good to have a color bar/legend or color by gymnosperm/angiosperm

RESPONSE: Yes, the colors on the figure 4 represent different species. The colorbar represents same species as on figure 6.

Lines 144 & section: I am all in favor of bold titles that help the reader get the main point, but somewhere in this section, I think you need to remind readers that this is assuming that species hydraulic traits are constant over time. You included it in the methods, which is helpful, but you should state that “assuming hydraulic traits for species are relatively constant within the same trees/population over time...” there is a weakening effect of hydraulic traits on DTR-growth loss relationship. This is a fair

assumption and its also what we have data for, but it's also possible that there is variation around p50 p88 values for different populations of species, and that these values could change over time (though, probably not the timescale of interest here).

RESPONSE: We thank the reviewer for this good suggestion. We have added the sentence: ‘we assume that hydraulic traits for species are relatively constant within the same trees/population over time’.

Fig S8 is compelling, but all the white circles are hard to see and imply that no correlation was done. Can you add a black outline to better visualize the size of the correlation?

RESPONSE: We have added a black outline for each circle in Fig. S8 accordingly.

Line 149 Become increasingly what?

RESPONSE: Become increasingly positive. We have added the missing word ‘positive’ accordingly.

Line 156 Influences? Instead of influence

RESPONSE: The word ‘influence’ was revised to ‘influences’ accordingly.

Line 169 DTR exerts* should this be plural here?

RESPONSE: The word ‘exert’ was changed to ‘exerts’ accordingly.

Line 203: Mitigates should probably be mitigated here?

RESPONSE: The word ‘mitigates’ was changed to ‘mitigated’ accordingly.

Paragraph starting at line 188: I appreciate the wide discussion of possible causes in this paragraph! It gives the discussion a lot of context.

RESPONSE: We thank the reviewer for the positive comments.

Lines 208-209 & throughout: Minor comment: There are a few places where past tense is used to describe results found in other studies, but unless this was shown with this data or in this study I think it needs a qualifier or description of where it was found. For example here, I'd change it to “was enhanced in other studies” or some qualifier.

RESPONSE: Thanks for your suggestion, we have implemented it over this paragraph, including changing the mentioned statement to ‘was enhanced in other studies’.

Line 258: Should this be “have”

RESPONSE: The word ‘having’ was replaced by ‘have’ accordingly.

Line 265-266: Could you also say that physiological site/tree level responses may differ from inferred community level responses?

RESPONSE: Yes. We have changed the conclusion sentence to ‘increasing asymmetric diurnal warming may affect species distributions and community dynamics, especially under changing drought regimes’ to show that physiological site/tree level responses may differ from inferred community level responses accordingly.

Line 292: And north American trees too?

RESPONSE: Yes. We have revised the sentence to ‘Our data was also clearly shifted towards northern European and north American forests’, accordingly.

Line 314: obtained*

RESPONSE: The word ‘obtain’ was changed to ‘obtained’ accordingly.

Line 317: I think this should be “correlations....were calculated”, because it was multiple correlations for different species?

RESPONSE: The sentence was revised to ‘Correlations between site tree-ring chronologies and site climate (i.e., temperature and precipitation) were calculated to detect the dominant climate factor for each site’ accordingly.

Line 327: Won’t the mean tree ring index be close to 1 for all chronologies?

RESPONSE: Yes, the reviewer is correct, the mean tree ring index is close to 1 for all chronologies. We calculated the mean tree ring indices during dry years and wet years to compare their differences between dry years and wet years. To make this point clearer, we have changed the sentence to “To investigate the variation of mean tree growth indices with site mean DTR during dry and wet years, across sites and species, we calculated correlations between site mean DTR and mean tree ring indices during dry years and wet years”.

Line 335-337: I know this was a decision that was probably made a long time ago, and I think you have enough in this manuscript, so I don’t think you should change it, but it would be *interesting* to see how the severe drought years relate to DTR. Maybe there is another manuscript in the works that digs into this?

RESPONSE: Yes. We had to consider whether we retain severe drought years or not. We have detected how the severe drought years relate to DTR for larch (Zhang et al. 2022, Science of the Total Environment), and found that reduced DTR cannot alleviate drought stress in severe drought years. Although we did not include an analyses of severe drought years in this study, we had found that adding severe drought years did not change our results. We agree with the reviewer also that it would be very interesting in the future to continue studying the mechanisms by which reduced DTR does (or does not) alleviate drought stress during severe drought years and we will continue working in this line of work in the coming years.

Line 353: Random selection with replacement or no replacement?

RESPONSE: With replacement. We had added the term accordingly.

Fig 6: This version looks great! Could you just add text explaining what the grey and white parts represent on the map

RESPONSE: Thank you for spotting this. The grey regions represent forest regions, and the white regions represent non-forest regions. We have added this to the caption of figure 6.

Note on figs S2-S3 & S8 the white colors don’t show up on the white background—could you make these a different color? Light grey? Or could you make the country backgrounds a different color? For figure S8, you could add a grey outline to the circles so you can still see the size of the white circles.

RESPONSE: We thank the reviewer for this good suggestion. We have added grey outlines for all the circles for all these figures.

Reviewer #2 (Remarks to the Author):

I think the authors did a commendable job addressing both mine and the other reviewer's comments. I do not have any additional concerns and recommend publishing this manuscript.

RESPONSE: We thank the reviewer for the supportive comments.